**Subject Category:**
Biology (whole organism)

behaviour/ecology

Cuvier's beaked whale, satellite telemetry, diving behaviour

**Author for correspondence:**
Jeanne M. Shearer
e-mail: jeanne.shearer@duke.edu

# Diving behaviour of Cuvier's beaked whales (*Ziphius cavirostris*) off Cape Hatteras, North Carolina

Jeanne M. Shearer[1], Nicola J. Quick[2], William R. Cioffi[1], Robin W. Baird[3], Daniel L. Webster[3], Heather J. Foley[2], Zachary T. Swaim[2], Danielle M. Waples[2], Joel T. Bell[4] and Andrew J. Read[2]

[1]Duke Marine Lab, University Program in Ecology, 135 Duke Marine Lab Rd, Beaufort, NC 28516, USA
[2]Duke University Marine Lab, 135 Duke Marine Lab Rd, Beaufort, NC 28516, USA
[3]Cascadia Research Collective, 218 $\frac{1}{2}$ W 4th Ave, Olympia, WA 98501, USA
[4]Environmental Conservation – Marine Resources Section (EV53), Naval Facilities Engineering Command Atlantic, Norfolk, VA 23508, USA

 JMS, 0000-0002-7784-870X; NJQ, 0000-0003-3840-6711;
WRC, 0000-0003-1182-8578

Cuvier's beaked whales exhibit exceptionally long and deep foraging dives. The species is little studied due to their deep-water, offshore distribution and limited time spent at the surface. We used LIMPET satellite tags to study the diving behaviour of Cuvier's beaked whales off Cape Hatteras, North Carolina from 2014 to 2016. We deployed 11 tags, recording 3242 h of behaviour data, encompassing 5926 dives. Dive types were highly bimodal; deep dives (greater than 800 m, $n = 1408$) had a median depth of 1456 m and median duration of 58.9 min; shallow dives (50–800 m, $n = 4518$) were to median depths of 280 m with a median duration of 18.7 min. Most surface intervals were very short (median 2.2 min), but all animals occasionally performed extended surface intervals. We found no diel differences in dive depth or the percentage of time spent deep diving, but whales spent significantly more time near the surface at night. Other populations of this species exhibit similar dive patterns, but with regional differences in depth, duration and inter-dive intervals. Satellite-linked tags allow for the collection of long periods of dive records, including the occurrence of anomalous behaviours, bringing new insights into the lives of these deep divers.

# 1. Introduction

Beaked whales (family Ziphiidae) are a diverse, but poorly understood, family of deep-diving cetaceans. These species spend little time at the surface and routinely perform foraging dives of more than an hour [1–5]. Beaked whales are typically found near the shelf break, over the continental slope [6], near oceanic islands with steep slopes [1,7], or in the vicinity of canyons and seamounts [8–10]. Most studies of their diet have been limited to small samples from stranded animals, but they appear to feed on a variety of cephalopods [8,11–13] and small fish [14], including both bathypelagic and mesopelagic prey [1,15].

The few species of beaked whales for which detailed dive records exist (Cuvier's beaked whales (*Ziphius cavirostris*), Blainville's beaked whales (*Mesoplodon densirostris*) and northern bottlenose whales (*Hyperoodon ampullatus*)) exhibit a stereotypic diving pattern, consisting of a single deep foraging dive (routinely deeper than 1000 m), followed by a series of relatively shallow dives [2,4,16–18]. The longest (137.5 min) and deepest (2992 m) dives of any mammal were recorded from Cuvier's beaked whales (*Z. cavirostris*) off southern California [3], although it is possible that these animals were exposed to Navy sonar during those deployments [3,19,20]. Echolocation clicks used in foraging are produced only below 200 m; whales are typically silent during shallow dives [1,21–23]. The shallow dives were originally believed to help prevent nitrogen gas formation in tissues [24], but occasional extended surface durations, and the ascent and descent rates of shallow dives, suggest that this is not the case [4,25]. Their silence during shallow dives may reduce the risk of predation from killer whales (*Orcinus orca*) [4,26], which are known to prey on beaked whales [27,28].

Despite long dives, Cuvier's beaked whales have short surface intervals, typically between 2 and 8 min [3,4,18]. Nevertheless, they occasionally exhibit extended surface durations of well over an hour, during which they remain within 40 or 50 m of the surface [2–4]. The duration of deep foraging dives is not correlated with the duration of the following surface interval [18]. However, the final surface intervals prior to deep dives are longer than surface intervals following a deep dive [3]. This indicates that Cuvier's beaked whales plan their dives and prepare for long dives by spending more time at the surface prior to diving, but, somewhat surprisingly, they do not compensate for deep dives with long recovery periods. Cuvier's beaked whales show little diel variation in deep-diving behaviour, with similar average deep dive depths and durations [2,3]. However, they spend more time near the surface at night, with fewer dives to intermediate depths [2,3].

Cuvier's beaked whales occur year-round off Cape Hatteras, North Carolina, where they occur in very high densities [29–31] and exhibit strong site fidelity [7,32,33]. This area has complex topography characterized by a steep shelf break and multiple deep canyons, over which the Labrador and Gulf Stream currents converge. We have deployed satellite tags on Cuvier's beaked whales in this location since 2014 as part of a long-term monitoring project for the U.S. Navy. Here, we describe the first record of the diving behaviour of Cuvier's beaked whales in the U.S. Atlantic. These data address a gap in knowledge for this species occurring in an area with multiple human uses, from occasional military exercises to extensive fishing and shipping traffic, and the recent potential for offshore seismic surveys [34].

# 2. Material and methods

## 2.1. Field methods

Between 2014 and 2016, we deployed satellite-linked depth-recording SPLASH10-292 tags (Wildlife Computers, Redmond, WA) in the low impact minimally percutaneous external-electronics transmitter (LIMPET) configuration [35] on or near the dorsal fins of Cuvier's beaked whales off Cape Hatteras, North Carolina (figure 1). Nine tags were in an extended depth configuration (calibrated to 2000 m with a 1 m resolution, with a pressure sensor designed not to exceed 3000 m). Two additional tags (ZcTag046 and ZcTag047) were in a standard depth configuration (calibrated to 1000 m with a 0.5 m resolution, with a pressure sensor designed to survive depths of 2000 m). We deployed the tags from a 9 m rigid-hulled boat, using a pneumatic rifle (DAN-INJECT JM 25, DanWild LLC, Austin, TX). Tags were attached by two surgical-grade titanium darts with backward-facing petals. Prior to tagging, we attempted to photograph the dorsal fin and body of all individuals in the group for identification and sex classification. From these photographs, we identified tagged animals using dorsal fin notches and patterns of scars on the body. Photos were compared with a catalogue of known individuals from the study area to aid in designating sex and age classes for individuals with

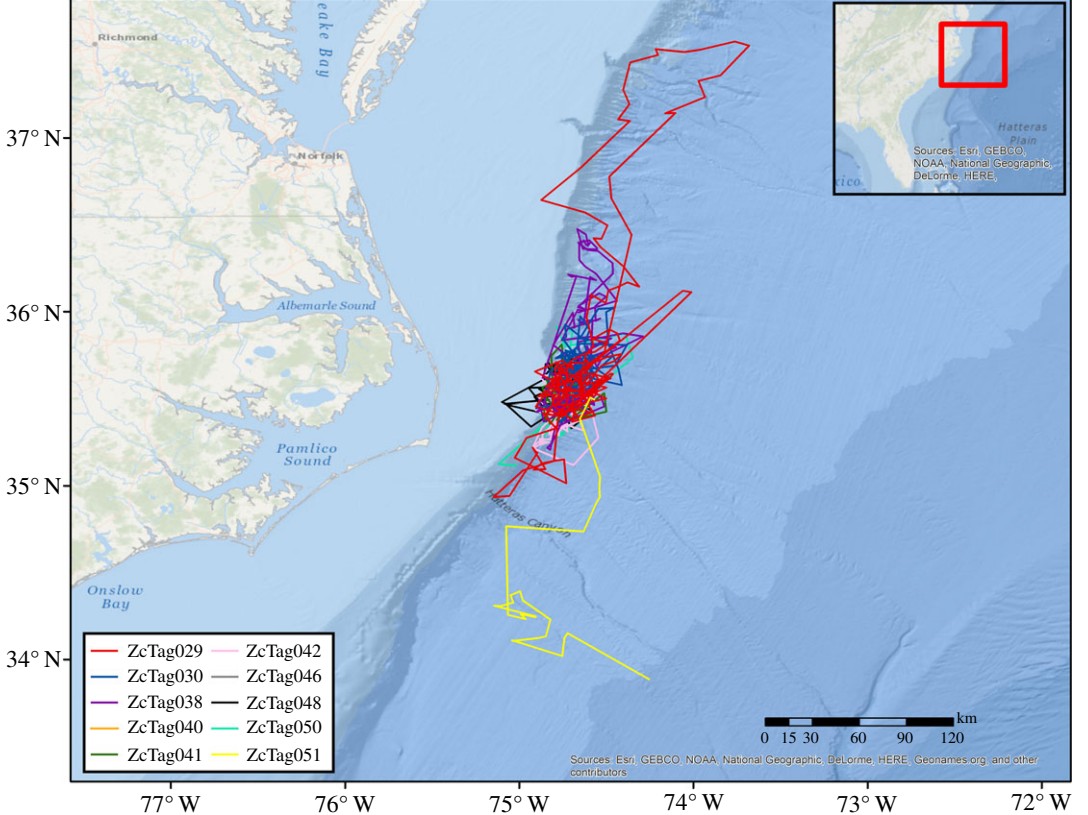

**Figure 1.** Study location near Cape Hatteras, North Carolina and movement tracks of tagged Cuvier's beaked whales. Animal locations were estimated from Argos positions (least-squares method) and filtered using the Douglas Argos filter (v. 8.5), following previous protocols [36].

a prior or subsequent sighting record. We classified whales as adult males if erupted teeth were visible at the end of their rostra or if they had extensive linear scarring and white pigmentation on their heads and bodies [28,37–39]. We classified whales as adult females if they were observed with small calves surfacing in close proximity or, if a calf was not present, based on relatively large body size and a lack of erupted teeth and extensive linear scarring [26,31]. Without the presence of a calf, it is difficult to distinguish adult females from subadult males. All classifications were made independently by two laboratories, and sighting histories and videos of tagging attempts for some of the animals increase our confidence in our classifications. Our sample of tagged individuals is biased towards males because the distinctive white pigmentation makes them more visible to observers and easier to track during tagging attempts. The white pigmentation and linear scarring in males increases our confidence in their classification. In addition, we avoided females with dependent calves due to conditions of our permits.

## 2.2. Analysis methods

### 2.2.1. Satellite tag programming

The satellite tags recorded and transmitted a series of dive statistics to polar-orbiting satellites via the Argos satellite system. We programmed the tags to transmit for 20 h per day. Depending on the year of deployment, tags were programmed to transmit daily for 25–28 days, followed by a duty cycle of every second or third day to maximize temporal coverage. Dive statistics included the start and end time of each dive, maximum depth and duration, as well as the duration of surface intervals, defined as the time elapsed between dives exceeding 50 m depth. Each surfacing period started when the conductivity sensor on the tag registered that the animal reached the surface after completing a qualifying dive. Maximum dive depths are recorded as two values, between which lies the true maximum value of the dive; we report the average of these two values. In addition, we programmed

**Table 1.** Data errors from Cuvier's beaked whale satellite tag deployments off Cape Hatteras, North Carolina.

| tag | error/anomaly | solution |
|---|---|---|
| ZcTag029 | One satellite message extended for 13 days and overlapped with other behaviour log messages. It included a dive of 230 h and a surface interval of 62 h. This message was probably corrupt and was erroneously retained. | The corrupt message was removed from data. Overlapping, normal behaviour log messages were kept. |
| ZcTag040 | Final two dives had speeds over 15 m s$^{-1}$. Probably reflects battery or pressure sensor failure just before the tag stopped transmitting. | The final message was removed. |
| ZcTag042 | Pressure sensor failed on 9 Nov 2015. Biologically impossible speeds occurred and pressure sensor offset indicated failure. | All data after 7 Nov 2015 were removed. |
| ZcTag047 | Pressure sensor failed on or before 27 May 2016 (2 days into deployment). Pressure sensor offset indicated failure. | Data from this tag were not used. |
| ZcTag050 | Low tag deployment resulted in possible under sampling of conductivity sensor. One-third of all surfacings were 2 s long. | Only depth data were used, all duration data were excluded. |
| ZcTag051 | Dive recorded to 3568 m, deeper than the rating of the tag. | This dive was removed from statistics; all other dives were retained. |

the tags to collect time-series data, in which depth was sampled every 2.5 min for a period of several hours, duty cycled throughout the tag duration.

### 2.2.2. Error checks

We checked all tag records systematically for errors that indicated failure or drift in any of the tag sensors. We reviewed the status messages provided by the tag to ensure that all sensors were operating as intended; this included information about battery voltage, the conductivity sensor and the pressure sensor's depth offset at the surface. In addition, we checked for biologically implausible records, including depth/duration combinations that required unrealistically high speeds (e.g. one tag record included two dives that would have required average speeds of over 15 m s$^{-1}$). We checked for message overlaps, which are usually corrected by the Argos system, but occurred on one tag. We used the time-series data as an external check of the maximum dive depth reported in the dive summaries for periods in which the series and behaviour data streams overlapped. We cross-checked every long surface period and back-to-back deep dives with the available series data stream. Gaps in the data record resulted from the duty cycle and from periods when the Argos system failed to receive messages due to poor satellite coverage or the surfacing behaviour of the whale. These gaps resulted in non-continuous data records. We omitted these gaps from all calculated statistics (e.g. inter-deep dive intervals were not calculated when a gap occurred between deep dives, and surface duration statistics were not calculated when a gap occurred immediately before or after a dive).

Table 1 presents a complete summary of tag errors and solutions. Of particular note, ZcTag050 had a high number (35%) of two-second surfacing durations. The tag was low on the body of this animal, and we suspect that this placement prevented the conductivity sensor from getting dry on many surfacings, therefore undersampling surface durations. We cannot determine whether this error should be attributed to the previous or following dive, or whether some shallow dives were missed entirely. We therefore decided to exclude all duration data from this animal, and we included this tag only in depth calculations. In addition, ZcTag051 recorded a dive to 3568 m, which is deeper than the previous record for this species. However, SPLASH10-292 tags have only been tested to 3000 m [3], so we chose to exclude this dive from all statistics reported. The tag gave no indication of failure, so we retained all other dives in our analysis.

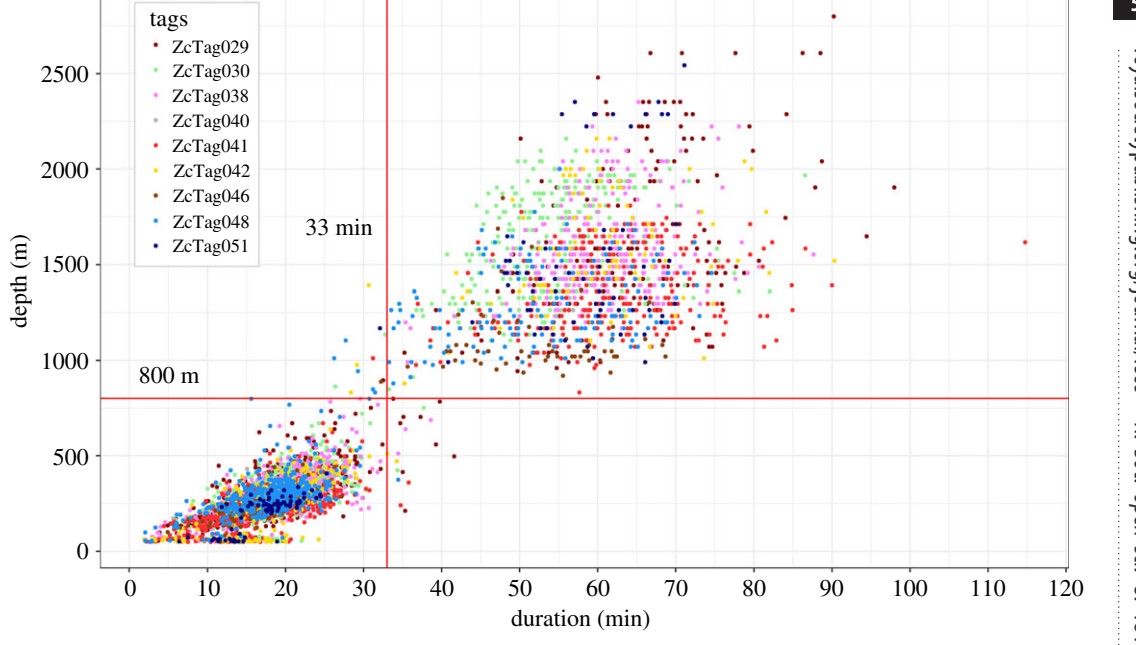

**Figure 2.** Cuvier's beaked whale dive depths plotted against dive durations. Dives were divided into two classes using a cut-off of 800 m as has been done previously [2,18]. Similar results are obtained with a 33 min cut-off.

### 2.2.3. Dive and surface classification

Dive depths were highly bimodal, with long, deep dives interspersed with a number of short and shallow dives (figure 2; electronic supplementary material, figure S1). Based on this bimodal distribution, we defined a *deep dive* to be any submergence greater than 800 m. Both a histogram of dive depths and a log survival plot indicated that this was a robust cut-off. We classified *surface intervals* according to type, based on Schorr *et al.* [3]. *Terminal surfacings* occurred just before a deep dive; *first surfacings* immediately followed a deep dive; *intermediate surface intervals* occurred between sequential shallow dives and *single surfacings* occurred between back-to-back deep dives. Surface intervals in which there were data gaps prior to or following the surface interval were not classified. We classified dives and surface intervals as occurring during day or night using a single point in the vicinity of the location of tag deployments as a reference (35°N, 75°W). We calculated sunrise and sunset times for each day from the NOAA solar calculator using the sunrise.set tool in the StreamMetabolism package in R [40,41]. We considered any event beginning after sunrise and prior to sunset to have occurred during the day, regardless of the time the event ended.

### 2.2.4. Calculations

We calculated individual and group medians, means and coefficients of variation for all dive and surface types. We provide both medians and means to illustrate the skewed nature of the data and to facilitate comparisons with other studies that used varying metrics. We calculated *inter-deep dive intervals* (IDDIs) as both the duration in minutes and the number of shallow dives between deep dives. We only calculated IDDIs for intervals in which there were no gaps in the data record between successive deep dives. Deep and shallow dive rates were calculated using the overall number of dives in each category divided by the total number of hours of data collected (i.e. sum of dive durations and surfacing periods) per animal. We calculated the per cent of time the animals spent in each event type (deep dives, shallow dives and surface intervals). We could not calculate the per cent of time in each depth layer due to the manner in which data were summarized before being transmitted.

### 2.3. Statistical methods

We used paired *t*-tests to test for diel differences in the median values for dive depth and duration, surface duration, IDDI and per cent of time spent in each event type. We employed a Kruskal–Wallis

multiple comparison test followed by a *post hoc* Dunn test to test for differences in duration between the four surface types.

# 3. Results

## 3.1. Summary of data

Between 2014 and 2016, we deployed 11 SPLASH10-292 satellite tags on Cuvier's beaked whales in the Cape Hatteras region (figure 1), collecting a total of 3242 h of data (table 2). Most (six) of the tagged whales were adult males, one was an adult female, two were probably adult females and two were probably adult males. Deployments ranged from 1 to 60 days, with a median of 28 days. We recorded a total of 5926 dives, ranging from 30 to 1517 dives per animal and 9 to 324 deep dives per animal. All animals performed a stereotypical pattern of a deep dive followed by a series of shorter, shallow dives (figure 3). The longest dive lasted 114.7 min, and the deepest dive was to 2800 m.

## 3.2. Dive behaviour

On average, animals performed almost three times as many shallow dives as deep dives. Deep dives ($n = 1408$) had an overall median depth of 1456 m and a median duration of 58.9 min (table 3). Median deep dive depths for individual animals ranged from 1048 to 1616 m and 51.6 to 64.5 min (table 3 and figure 4). Considering all animals, deep dives ranged from 832 to 2800 m depth and from 26.2 to 114.7 min in duration. Shallow dives ($n = 4518$) had an overall median depth of 280 m (individual medians from 224 to 320 m) and 18.7 min (individual medians 16.1 to 19.4 min) (table 3 and figure 5).

Deep dives occurred at a rate of 0.43 dives per hour; shallow dives occurred at 1.40 dives per hour (table 4). Inter-deep dive intervals lasted for an overall median of 74.6 min, ranging from 38.5 to 89.5 min per individual (table 4). All animals had at least one IDDI of over 100 min, and two whales had IDDIs of over 300 min, indicating very long periods between foraging dives. A median of three shallow dives occurred between deep dives, but individuals sometimes performed up to 14 shallow dives between deep dives. All animals except ZcTag040 (the shortest tag record) performed back-to-back deep dives, although these were infrequent, and comprised only 1% of all inter-deep dive intervals.

## 3.3. Surface (less than 50 m depth) behaviour

Despite performing very deep and long dives, Cuvier's beaked whales spent remarkably little time at or near the surface, with a median surface duration of only 2.2 min (table 5). Individual medians for all animals except ZcTag051 ranged from 1.8 to 2.3 min; ZcTag051 was an outlier with a median of 4.2 min. Despite an overall median of only 2.2 min, all animals exhibited occasional extended surface durations of over 30 min (2.7% of all surface intervals). The maximum surface duration for any animal was 310 min. Surface durations varied considerably based on their type (table 5 and figures 6 and 7). The longest surface durations were single surfacings between back-to-back deep dives, with a median of 20 min (range 2.2–310 min). Terminal surfacings had a median of 4.0 min (range 36 s to 140 min); first surfacings had a median of 2.3 min (range 2 s to 129 min). Finally, intermediate surfacings were the shortest at a median of 1.9 min (range 2 s to 151 min). Surface durations of 2 s ($n = 12$, 0.2%) were probably under-representations due to the conductivity sensor occasionally not getting dry at the surface (see Material and methods: Error checks for more detail).

The result of the Kruskal–Wallis test was highly significant ($p < 0.001$), and *post hoc* Dunn tests showed that all categories were significantly different from each other. Single surfacings were significantly longer than terminal ($p < 0.001$), first ($p < 0.001$) and intermediate ($p < 0.001$). Terminal surfacings were significantly longer than first ($p < 0.001$) and intermediate ($p < 0.001$). First surfacings were significantly longer than intermediate ($p < 0.001$).

## 3.4. Per cent time in dive type

The whales spent, on average, the same proportion of time in deep and shallow dives (deep dives: 43%, shallow dives 43%, table 6); approximately 14% of their time was spent during surface periods (i.e. periods with no dives greater than 50 m).

**Table 2.** Summary of Cuvier's beaked whale tag deployments near Cape Hatteras. Deployment duration is given in the number of days between the tag deployment and tag failure or cessation of data transmission. The number of hours of behaviour data excludes gaps and includes only recorded data, not tag life. Deep dives were considered as any dive greater than 800 m depth; shallow dives were less than 800 m depth. Age and sex were determined from photographs taken at the time of deployment within 2 days of deployment and the data were not used in this analysis.

| tag ID | date tagged | deployment duration days | behaviour data hours (days) | number of deep (>800 m) dives | number of shallow (<800 m) dives | age/sex |
|---|---|---|---|---|---|---|
| ZcTag029 | 13 May 2014 | 59.7 | 437.2 (18.2) | 157 | 653 | probably adult female |
| ZcTag030 | 16 Sep 2014 | 38.5 | 616.4 (25.7) | 260 | 943 | adult male |
| ZcTag038 | 14 June 2015 | 56.4 | 825.7 (34.4) | 324 | 1193 | adult male |
| ZcTag040 | 14 June 2015 | 0.7 | 17.2 (0.7) | 9 | 21 | adult male |
| ZcTag041 | 15 Oct 2015 | 32.3 | 559.0 (23.3) | 274 | 695 | probably adult male |
| ZcTag042 | 21 Oct 2015 | 17.4 | 224.0 (9.3) | 98 | 325 | adult male |
| ZcTag046 | 25 May 2016 | 10.0 | 97.2 (4.1) | 59 | 125 | adult male |
| ZcTag047 | 25 May 2016 | ~2 | 0 (0) | 0 | 0 | adult female |
| ZcTag048 | 27 May 2016 | 30.3 | 286.3 (11.9) | 144 | 439 | probably adult male |
| ZcTag050 | 20 Aug 2016 | 24.5 | 39.0 (1.6) | 17 | 44 | probably adult female |
| ZcTag051 | 21 Aug 2016 | 9.3 | 138.8 (5.8) | 66 | 80 | adult male |
| Total | | 281.1 | 3242.1 (135.1) | 1408 | 4518 | |

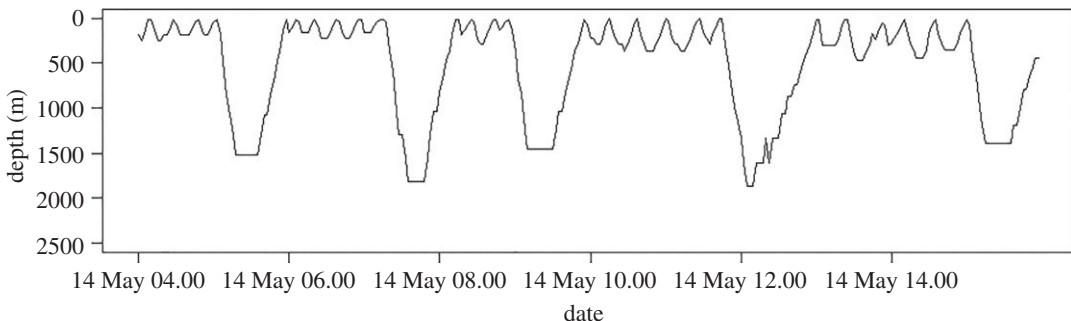

**Figure 3.** Time series of depths (recorded every 2.5 min) over 12 h for ZcTag029, showing deep and shallow dives.

## 3.5. Diel differences

Deep dives (using medians) were significantly longer during the day, and shallow dives were both significantly longer and deeper, and occurred at a significantly higher rate, during the day. There was no diel difference in the duration of inter-deep dive intervals. For each animal, the maximum surface duration was significantly longer at night than during the day. Both terminal and single surface intervals were significantly longer at night than during the day (table 7 and figure 8). There was no significant difference in the per cent of time spent in deep dives during the day versus at night (table 7 and figure 9). However, significantly more time was spent in shallow dives during the day than at night and significantly more time was spent at the surface at night (table 7 and figure 9).

# 4. Discussion

## 4.1. Extreme diving

The results of our study confirm that the Cuvier's beaked whales found off Cape Hatteras, North Carolina demonstrate remarkable diving capabilities, similar to findings in other areas for this species [2–5,18]. The whales we tagged routinely made deep dives to depths of greater than 1500 m, lasting for an hour or more. Counterintuitively, these deep dives were not associated with prolonged surface durations, either preceding or following the submergence. These routinely short surfacings raise additional questions about other aspects of their behaviour. If beaked whales rarely spend time at the surface, when do they socialize or rest? Do the whales socialize and rest during deep dives or during the shallow dives interspersed between these deep dives? Further studies, possibly employing advanced telemetry approaches, are required to determine when behavioural states other than foraging occur.

## 4.2. Comparisons to other studies

Cuvier's beaked whales have been tagged in several other locations, enabling some preliminary comparisons to be drawn among populations (table 8). Long-term satellite tags have been deployed in southern California [3], the Bahamas [5] and Hawaii [42], and these studies have provided the most comparable data to the records we present here. Suction-cup data loggers have been deployed in the Ligurian Sea, Italy [4] and in Hawaii [2,18], but these tags are of relatively short duration, although with considerably higher resolution. The Cuvier's beaked whales off Cape Hatteras exhibited dive patterns similar to those of other populations, but with some differences in dive depths, durations and inter-deep dive intervals. The median depth of deep dives of Cuvier's beaked whales near Cape Hatteras was similar to that observed in southern California, but greater than that in Italy, Hawaii and the Bahamas (table 8). Despite these deeper dive depths, the duration of deep dives off Cape Hatteras was shorter than those in southern California, Hawaii and the Bahamas, although similar to those in Italy (table 8). Therefore, our animals dived to deeper average depths but for shorter durations than in other areas. Very little is known about the prey fields for any population of beaked whale, but comparisons of the density and availability of deep-water prey among sites could inform these foraging behaviour differences.

**Table 3.** Summary of dive depths and durations for Cuvier's beaked whales. Medians and means are given for each individual, along with coefficients of variation for the mean (CV). The group median, mean and CV are given under 'total'. Depths are calculated in metres; durations are calculated in minutes. The maximum depth and duration for deep dives are also given. Durations were not calculated for ZcTag050 due to probable errors of undersampling surfacings.

| tag ID | deep dives | | | | | shallow dives | | | | |
| --- | --- | --- | --- | --- | --- | --- | --- | --- | --- | --- |
| | no. | depth median / mean (CV) | maximum | duration median / mean (CV) | maximum | no. | depth median / mean (CV) | maximum | duration median / mean (CV) | maximum |
| ZcTag029 | 157 | 1552 | 2800 | 64.5 | 98.0 | 653 | 280 | 800 | 18.5 | 41.6 |
| | | 1663 (25.7) | | 64.4 (16.8) | | | 297 (39.5) | | 18.5 (29.0) | |
| ZcTag030 | 260 | 1616 | 2160 | 53.8 | 86.6 | 943 | 320 | 752 | 19.4 | 37.7 |
| | | 1612 (16.7) | | 54.3 (15.5) | | | 315 (30.8) | | 19.4 (21.3) | |
| ZcTag038 | 324 | 1520 | 2352 | 60.5 | 87.6 | 1193 | 280 | 800 | 19.2 | 38.6 |
| | | 1566 (14.7) | | 60.8 (11.1) | | | 289 (30.0) | | 19.0 (23.1) | |
| ZcTag040 | 9 | 1424 | 1616 | 55.9 | 68.3 | 21 | 241 | 432 | 18.0 | 29 |
| | | 1399 (10.5) | | 56.8 (8.1) | | | 267 (32.8) | | 18.6 (27.3) | |
| ZcTag041 | 274 | 1296 | 1808 | 60.9 | 114.7 | 695 | 241 | 672 | 17.3 | 35.8 |
| | | 1355 (15.3) | | 61.7 (16.2) | | | 234 (38.2) | | 17.3 (28.7) | |
| ZcTag042 | 98 | 1520 | 2160 | 59.6 | 90.3 | 325 | 272 | 640 | 18.5 | 34.3 |
| | | 1554 (19.0) | | 59.4 (18.0) | | | 261 (44.7) | | 18.2 (28.0) | |
| ZcTag046 | 59 | 1048 | 1848 | 51.6 | 68.7 | 125 | 224 | 456 | 16.1 | 27.2 |
| | | 1109 (16.4) | | 52.2 (16.2) | | | 231 (30.5) | | 16.2 (25.2) | |
| ZcTag048 | 144 | 1232 | 2000 | 52.8 | 74.6 | 439 | 296 | 800 | 18.6 | 34.4 |
| | | 1284 (17.1) | | 52.7 (21.1) | | | 294 (33.9) | | 18.0 (25.7) | |
| ZcTag050 | 17 | 1616 | 2160 | — | — | 44 | 248 | 608 | — | — |
| | | 1614 (15.2) | | | | | 226 (58.9) | | | |
| ZcTag051 | 66 | 1424 | 2544 | 59.5 | 71.1 | 80 | 241 | 408 | 17.8 | 25.3 |
| | | 1507 (26.6) | | 58.1 (13.5) | | | 206 (44.8) | | 17.3 (23.0) | |
| total | 1408 | 1456 | 2800 | 58.9 | 114.7 | 4518 | 280 | 800 | 18.7 | 41.6 |
| | | 1492 (20.8) | | 58.7 (16.9) | | | 282 (36.2) | | 18.5 (25.5) | |

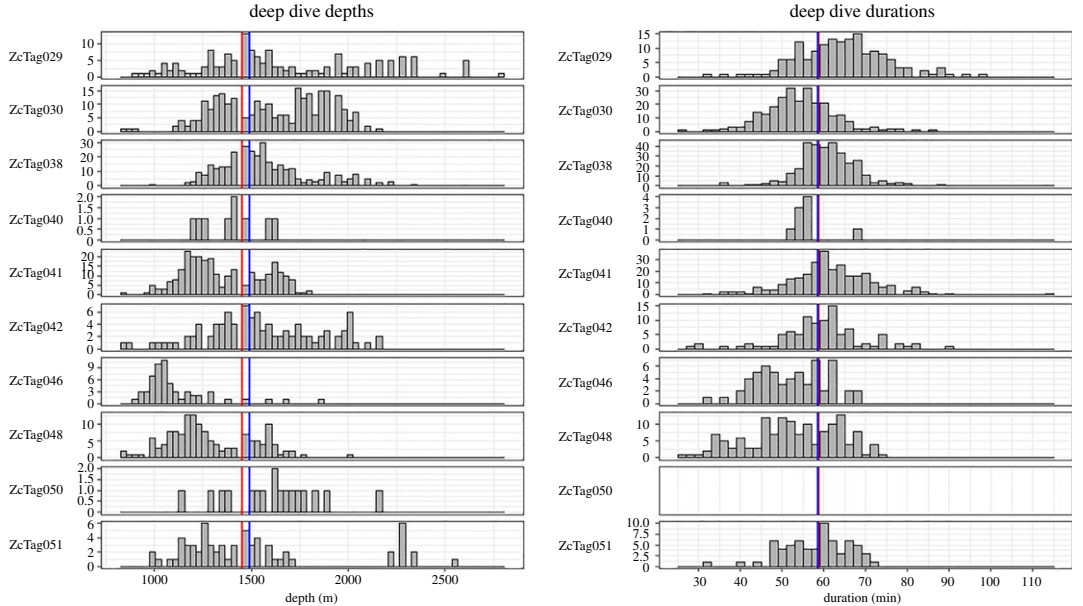

**Figure 4.** Histograms of deep dive depths and durations for individual Cuvier's beaked whales. The red vertical line is the median for all animals combined; the blue line is the overall mean. Durations were not calculated for ZcTag050 due to probable errors of undersampling surfacings. The *y*-axis scale varies between animals.

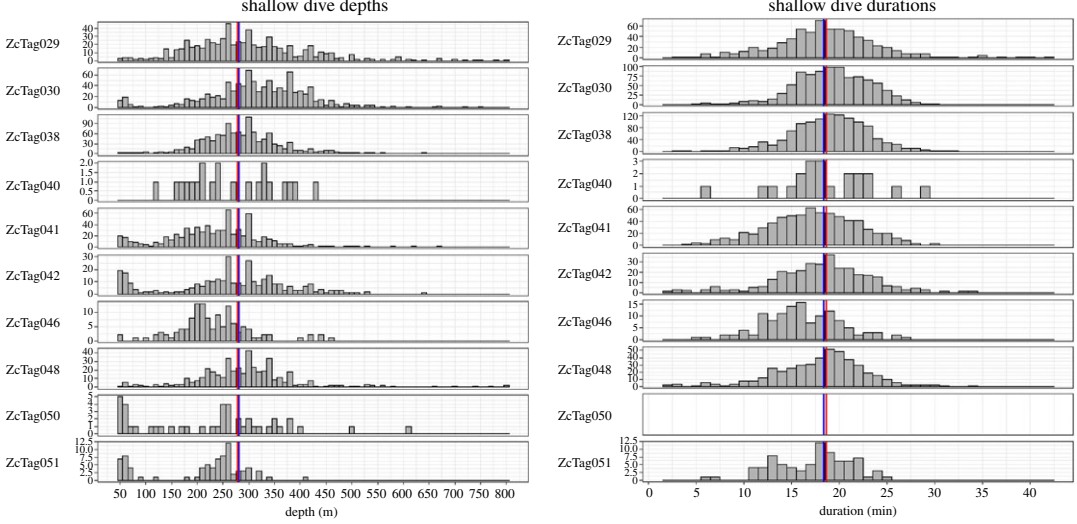

**Figure 5.** Histograms of shallow dive depths and durations for individual Cuvier's beaked whales. The red vertical line is the median for all animals combined; the blue line is the overall mean. Durations were not calculated for ZcTag050 due to probable errors of undersampling surfacings. The *y*-axis scale varies between animals.

Surface intervals of Cuvier's beaked whales were consistently short across all studies reporting this parameter (southern California and Italy) [3,4]. Despite typically short surface intervals, the animals we tagged all performed extended surface intervals, with a maximum of 310 min, and all animals experienced at least one surface duration lasting over 30 min. Extended surface durations have also been recorded in other areas (table 8). Thus, although relatively rare, it appears that Cuvier's beaked whales occasionally perform extended surface intervals, which have an unknown behavioural function.

These comparisons are limited due to different sample sizes and attachment durations, but we conclude that there are biologically meaningful differences in the diving behaviour of Cuvier's beaked whales in different locations, highlighting the importance of population-specific studies prior to making assessments of behaviour for monitoring or other purposes. In particular, the differences in foraging behaviour (seen in differences in maximum depths, durations and inter-deep dive intervals, as well as the rates of deep foraging dives) between locations may indicate differences in prey type,

**Table 4.** Inter-deep dive intervals and dive rates for Cuvier's beaked whales off Cape Hatteras. Medians and means are given for each individual, along with coefficients of variation for the mean (CV). The group median, mean and CV are given under 'total'. Inter-deep dive intervals were only calculated for periods with continuous behaviour records between deep dives; records with gaps were not used. Inter-deep dive intervals were calculated in both minutes (IDDI min) and as the number of shallow dives between deep dives (IDDI number); maximum values are given for both. Durations were not calculated for ZcTag050 due to probable errors of undersampling surfacings. Dive rates are calculated as the number of dives of each type divided by total hours of data recorded.

| tag ID | IDDI (min), median mean (CV) | IDDI (min) maximum | IDDI (number) median mean (CV) | IDDI (number) maximum | deep dive rate (dives h$^{-1}$) | shallow dive rate (dives h$^{-1}$) |
|---|---|---|---|---|---|---|
| ZcTag029 | 87.9 98.3 (53.9) | 310.0 | 4.0 3.7 (53.9) | 9 | 0.36 | 1.49 |
| ZcTag030 | 80.8 84.6 (50.0) | 222.9 | 3.5 3.5 (53.6) | 9 | 0.42 | 1.53 |
| ZcTag038 | 89.5 92.2 (39.7) | 337.3 | 4.0 3.7 (44.6) | 14 | 0.39 | 1.44 |
| ZcTag040 | 58.5 62.0 (45.8) | 107.1 | 2.0 2.5 (42.8) | 4 | 0.52 | 1.22 |
| ZcTag041 | 55.7 59.3 (49.7) | 202.9 | 2.0 2.5 (59.1) | 8 | 0.49 | 1.24 |
| ZcTag042 | 74.0 74.4 (48.0) | 168.3 | 3.0 3.0 (51.7) | 7 | 0.44 | 1.45 |
| ZcTag046 | 38.5 38.9 (67.1) | 100.3 | 2.0 1.7 (74.4) | 5 | 0.61 | 1.29 |
| ZcTag048 | 56.3 60.6 (66.5) | 209.0 | 3.0 2.8 (66.3) | 9 | 0.50 | 1.53 |
| ZcTag050 | — | — | — | — | — | — |
| ZcTag051 | 53.5 67.9 (59.2) | 182.5 | 1.0 1.1 (86.4) | 4 | 0.48 | 0.57 |
| total | 74.6 76.6 (53.7) | 337.3 | 3.0 3.0 (59.0) | 14 | 0.43 | 1.40 |

**Table 5.** Surface duration behaviour of Cuvier's beaked whales off Cape Hatteras. See the text for the definition of surface types. Surface durations were not calculated for ZcTag050 due to probable data errors. Medians and means are given for each individual, along with coefficients of variation for the mean (CV). The number of surfacings in each category is less than the total in the *all surfaces* column because some surfacings (4.2%) could not be classified due to data gaps.

| tag ID | all surface durations (min) | | | | terminal surface duration | | | first surface duration | | | intermediate surface duration | | | single surface duration | | |
|---|---|---|---|---|---|---|---|---|---|---|---|---|---|---|---|---|
| | n | median | mean (CV) | maximum | n | median | mean (CV) | n | median | mean (CV) | n | median | mean (CV) | n | median | mean (CV) |
| ZcTag029 | 807 | 2.3 | 5.0 (304.4) | 310.0 | 139 | 4.3 | 7.1 (145.3) | 149 | 2.5 | 5.6 (261.6) | 457 | 2.1 | 3.4 (259.5) | 4 | 31.2 | 93.6 (156.4) |
| ZcTag030 | 1204 | 2.2 | 3.8 (216.7) | 141.4 | 236 | 4.3 | 5.8 (110.8) | 237 | 2.7 | 4.1 (218.2) | 660 | 1.9 | 2.4 (163.6) | 11 | 10.7 | 44.4 (106.4) |
| ZcTag038 | 1519 | 2.2 | 4.8 (239.1) | 139.9 | 314 | 4.0 | 8.7 (196.9) | 318 | 2.3 | 3.5 (217.3) | 858 | 2.1 | 3.6 (245.6) | 4 | 10.8 | 29.2 (144.6) |
| ZcTag040 | 30 | 2.3 | 4.4 (140.1) | 32.6 | 8 | 8.1 | 10.4 (96.4) | 9 | 2.5 | 2.6 (13.8) | 12 | 2.0 | 2.0 (11.0) | 0 | — | |
| ZcTag041 | 967 | 2.0 | 4.7 (190.4) | 96.5 | 252 | 3.8 | 8.6 (137.3) | 249 | 1.7 | 2.8 (165.9) | 428 | 1.7 | 2.6 (159.4) | 17 | 30.8 | 34.6 (69.3) |
| ZcTag042 | 421 | 2.1 | 4.1 (181.2) | 85.4 | 88 | 4.2 | 5.2 (69.9) | 89 | 2.4 | 4.6 (219.0) | 211 | 1.8 | 3.2 (216.2) | 2 | 17.0 | 17.0 (103.7) |
| ZcTag046 | 187 | 1.8 | 3.9 (190.2) | 57.8 | 50 | 3.1 | 7.4 (153.4) | 51 | 1.6 | 2.2 (95.3) | 64 | 1.4 | 1.7 (117.5) | 7 | 3.5 | 10.5 (137.2) |
| ZcTag048 | 583 | 2.0 | 2.9 (109.6) | 40.8 | 130 | 3.4 | 4.9 (106.2) | 122 | 2.3 | 2.7 (80.5) | 288 | 1.8 | 1.9 (69.8) | 12 | 3.9 | 5.3 (64.3) |
| ZcTag050 | — | | | | — | | | — | | | — | | | — | | |
| ZcTag051 | 139 | 4.2 | 22.3 (155.3) | 182.5 | 45 | 13.4 | 20.0 (116.4) | 47 | 2.7 | 9.4 (223.8) | 29 | 2.4 | 11.6 (176.6) | 15 | 85.0 | 84.2 (52.4) |
| Total | 5857 | 2.2 | 4.7 (245.7) | 310 | 1262 | 4.0 | 7.7 (162.7) | 1271 | 2.3 | 3.9 (237.1) | 3007 | 1.9 | 3.0 (232.1) | 72 | 20 | 41.7 (125.3) |

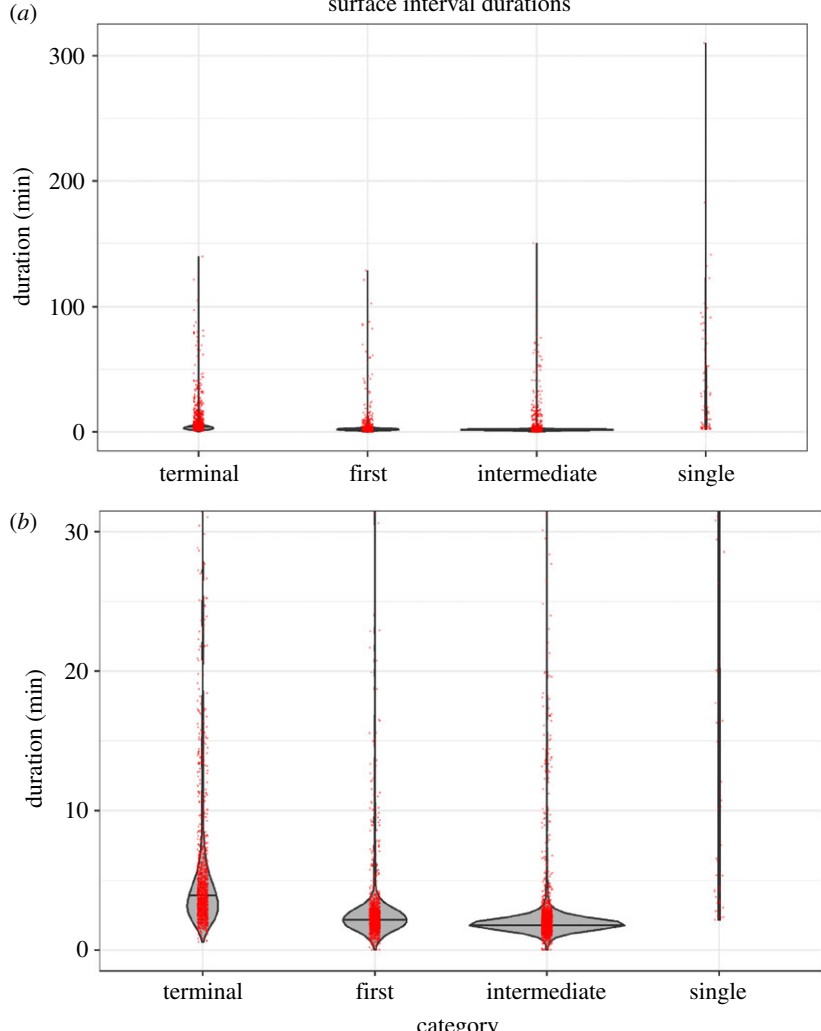

**Figure 6.** (*a*) Violin plot of all surface interval durations for Cuvier's beaked whales off Cape Hatteras by type, including outliers. (*b*) Violin plot of surface interval durations, cut-off at 30 min duration to improve readability of the plot. Violin plots show the distribution of points. Red dots are actual data points and are jittered for clarity.

density or behaviour [3,19,20]. Future studies including estimations of prey type and density and incorporating bathymetry and local oceanography could help tease apart these differences. In addition, our sample sizes are small and it is often difficult to sample both sexes equally. Increasing sample size and including information on the sex and age of the tagged animal (through increasing effort to obtain biopsy samples and ID photos) would help determine if there are age or sex class differences in behaviour in addition to population-level differences.

## 4.3. Diel differences

We observed no diel differences in depth or per cent of time spent in deep dives [3,5], although deep dives were significantly longer during the day (table 7). This, together with similar patterns of deep and shallow dives made during day and night, indicates that Cuvier's beaked whales do not exhibit diel variation in foraging behaviour. The whales do not appear to feed on deep scattering layer organisms, which would result in shallower foraging depths at night (e.g. [15,43,44]). We suspect that, at least off Cape Hatteras, Cuvier's beaked whales are foraging at or near the sea floor. Unfortunately, the resolution of the bathymetry coverage in this area, and the relatively large errors associated with location estimates generated via Argos, prohibits fine-scale analysis of the relationship between dive depth and bathymetry. Off the Bahamas, Cuvier's beaked whale deep dive depths are correlated with bathymetry, suggesting that the whales are foraging near the sea floor [5]. One other line of evidence for benthic foraging are gouges that have been observed in soft bottom sediments on seamounts and

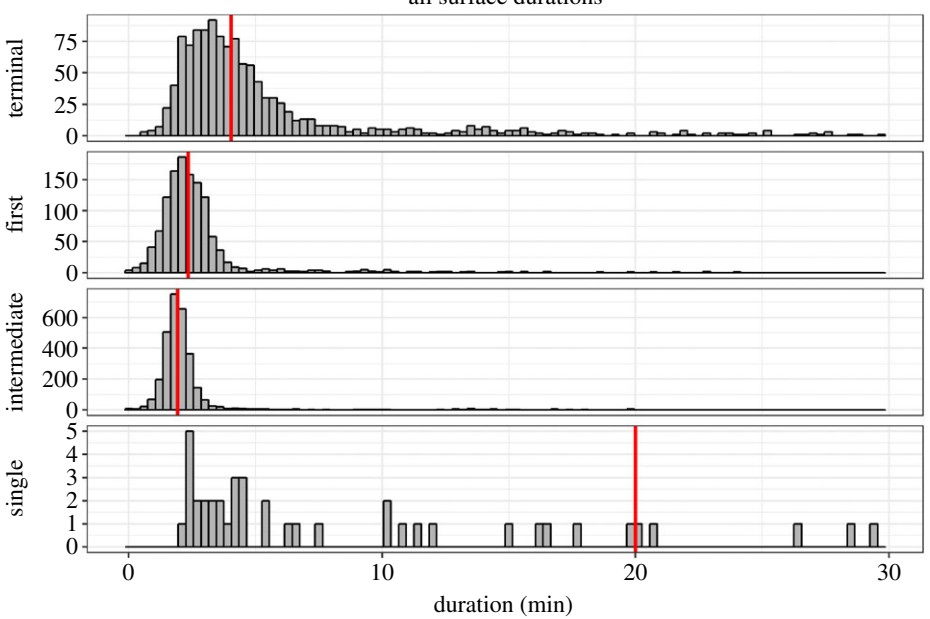

**Figure 7.** Histogram of surface durations for Cuvier's beaked whales off Cape Hatteras, by category. Surface durations longer than 30 min were removed for clarity (refer to figure 6a for outliers). Red vertical lines denote the median of each category.

**Table 6.** Per cent of time Cuvier's beaked whales spent in each dive category. Per cent of time in a category was not calculated for ZcTag050 due to possible errors in undersampling surface intervals.

| tag ID | dive category | | |
|---|---|---|---|
| | deep (greater than 800 m) | shallow (50–800 m) | surface (less than 50 m) periods |
| ZcTag029 | 38.5 | 46.2 | 15.3 |
| ZcTag030 | 38.2 | 49.4 | 12.5 |
| ZcTag038 | 39.8 | 45.7 | 14.6 |
| ZcTag040 | 49.4 | 37.8 | 12.8 |
| ZcTag041 | 50.4 | 35.9 | 13.7 |
| ZcTag042 | 43.3 | 44.0 | 12.7 |
| ZcTag046 | 52.8 | 34.7 | 12.5 |
| ZcTag048 | 44.2 | 46.1 | 9.7 |
| ZcTag050 | — | — | — |
| ZcTag051 | 46.1 | 16.6 | 37.3 |
| Total | 43 | 43 | 14 |

volcanoes, both in the North Atlantic and elsewhere, and interpreted as resulting from benthic prey capture attempts by Cuvier's beaked whales [45–47].

We did observe significant diel differences in shallow dives and surface behaviour (figures 8 and 9). The whales we tagged spent more time in the upper 50 m at night, and maximum, average terminal and average single surface durations were all significantly longer at night. Most extended surface intervals occurred at night: of 3166 daytime surfacings, only 14 were longer than 30 min, with the longest at 132 min (figure 8). By contrast, 143 of 2691 of night-time surface intervals were over 30 min, up to a maximum of 310 min. Longer maximum surface intervals at night were also observed from this species in Hawaii, where the longest daytime intervals were only 7 min, but the longest night-time surface intervals lasted up to 161 min [2]. Schorr *et al.* [3] also reported that most long (greater than 60 min) surfacings occurred at night and that whales spent significantly more time at the surface at night. Beaked whales spend little time at the surface, where they are typically silent, potentially to avoid detection by killer whales (*Orcinus orca*), which forage in near-surface waters [4,42]. Some beaked

**Table 7.** Diel differences in the behaviour of Cuvier's beaked whales using paired *t*-tests of the median values of each parameter. Deep dives are greater than 800 m, shallow dives are to depths of 50–800 m and surface intervals are between dives of 50 or more m. Paired *t*-tests are conducted on individual medians, but the overall median for all animals during the day and at night is displayed for each parameter to illustrate effect sizes. *p*-values less than 0.05 were considered significant.

| parameter | overall median | | *p*-value (paired *t*-test of individual medians) |
|---|---|---|---|
| | day | night | |
| deep dives | | | |
| depth | 1456 m | 1456 m | 0.570 |
| duration | 61.0 min | 56.4 min | 0.003** |
| dive rate | 0.41 h$^{-1}$ | 0.42 h$^{-1}$ | 0.096 |
| shallow dives | | | |
| depth | 303.5 m | 240.5 m | <0.001*** |
| duration | 20.1 min | 16.9 min | <0.001*** |
| dive rate | 1.49 h$^{-1}$ | 1.18 h$^{-1}$ | 0.009** |
| IDDI | | | |
| median duration | 84.2 min | 65.1 min | 0.130 |
| maximum duration | 203 min | 168 min | 0.439 |
| number shallow | 4 dives | 2 dives | <0.001*** |
| surface | | | |
| median duration (all types) | 2.1 min | 2.4 min | 0.321 |
| maximum duration | 15.0 min | 96.5 min | 0.005** |
| terminal duration | 3.4 min | 4.9 min | 0.039* |
| first duration | 2.2 min | 2.5 min | 0.254 |
| intermediate duration | 1.9 min | 2.0 min | 0.340 |
| single duration | 3.7 min | 30.1 min | 0.035* |
| per cent time in dive category | | | |
| time in deep dives | 41.9% | 43.2% | 0.463 |
| time in shallow dives | 49.4% | 36.1% | <0.001*** |
| time at surface | 8.7% | 20.7% | <0.001*** |

Significance level: \*$p = 0.05$, \*\*$p = 0.01$, \*\*\*$p < 0.001$.

whales begin clicking at shallower depths at night [1]. If these aspects of their behaviour represent a tactic to avoid detection by potential predators, then the fact that they spend significantly more time at the surface at night may also indicate a lower predation risk during periods of darkness [2]. Perhaps, mammal-eating killer whales are less of a threat to beaked whales at night than during the day [48,49].

## 4.4. Importance of long-term tags and individual variation

Cuvier's beaked whales exhibit similar patterns of deep dives interspersed with series of shallow dives in a variety of locations around the world. As a result, this species is sometimes assumed to perform this stereotypical behaviour across all individuals, seasons and locations. Beaked whale studies have often been limited by sample size. Some prior studies used short-duration time-depth recorders (TDRs) and DTAGs [2,4,18], which give detailed diving and movement behaviour, but limit the opportunity to observe lower frequency variation in behaviour and the occurrence of relatively rare behaviours. Our study confirms that different behavioural states emerge with longer duration tag deployments, including extended surface intervals and diel variations in surface behaviour that are unlikely to be captured by short-term tags [3]. Using a combination of short-term, high-resolution tags along with long-term tag deployments is ideal to capture both low-frequency behavioural states and detailed diving and movement behaviour [19,20]. In addition, previous DTAG and TDR data indicated little individual variation, but longer LIMPET deployments have revealed differences both between and

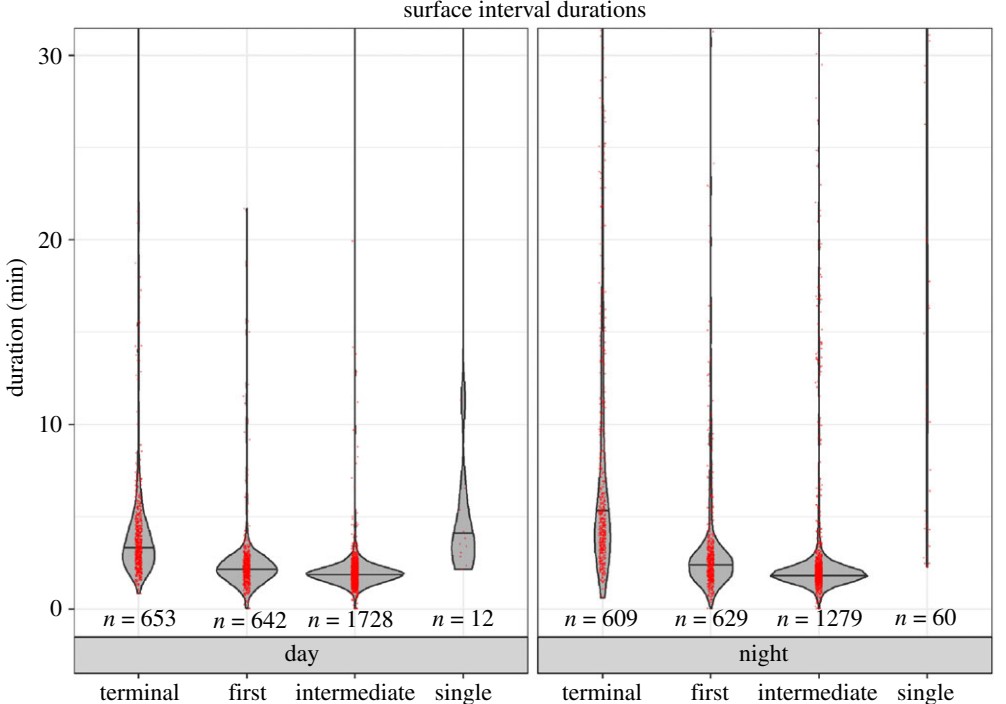

**Figure 8.** Violin plot of surface interval durations by category, split by time of day for Cuvier's beaked whales off Cape Hatteras. Violin plot shows distributions of data points. Red dots are actual data points, jittered for clarity. The number of observations in each category is given below each violin.

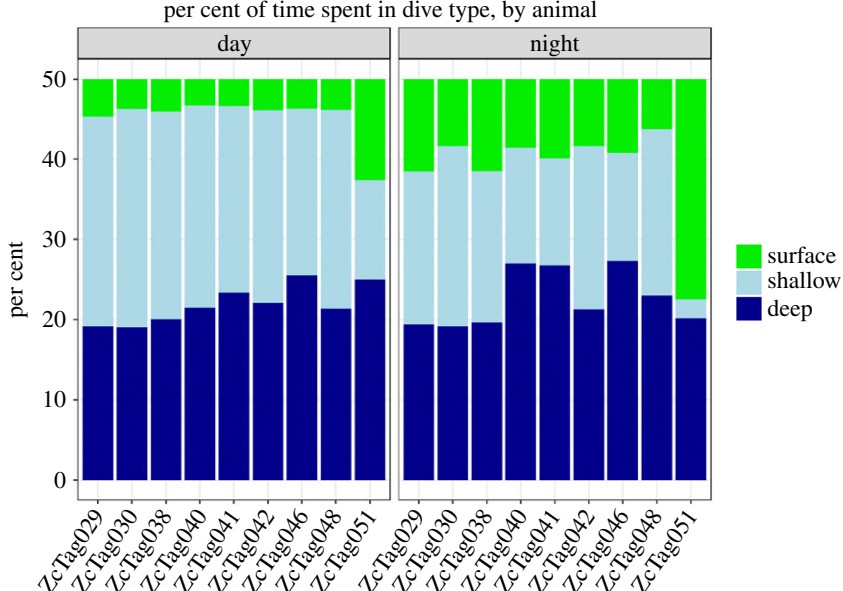

**Figure 9.** Per cent of time spent in each dive type, separated by individual and time of day for Cuvier's beaked whales off Cape Hatteras. ZcTag050 was not included due to probable errors in undersampling surface durations.

within individuals over time [3]. In this study, we did not quantitatively assess individual variation as data gaps meant that not all individuals were fully sampled. In the future, we aim to assess individual variation in Cuvier's beaked whales off Cape Hatteras using tags programmed to limit data gaps and provide more complete sampling of individuals.

## 4.5. Improvements to tag function

Studies of beaked whale diving behaviour are challenging because of their offshore distribution and limited time spent at the surface. In addition, their remarkable diving behaviour presents mechanical

**Table 8.** Comparison of dive characteristics reported here and from Cuvier's beaked whales tagged in other areas. All depths are in metres and all durations are in minutes. The method of calculating averages (mean, median) is stated with the number, when relevant. 'Mean' or 'median' indicate pooling of all individuals. 'Group mean' in Schorr et al. [3] indicates the mean of the individual medians. 'Grand mean' in Baird [42] indicates the mean of the individual means. Deep dives were defined as follows: this study, greater than 800 m; Schorr et al. [3]. K-means cluster analysis separating dive types; Tyack et al. [4] cluster analysis showing deep dives greater than 700 m; Joyce et al. [5], follows Tyack et al. [4] methods; Baird [42], greater than 800 m. n.a. indicates a metric was not given in the article.

| | this study | Schorr et al. [3] | Joyce et al. [5] | Tyack et al. [4] | Baird [42] |
|---|---|---|---|---|---|
| *study characteristics* | | | | | |
| tag type | Splash 10-292 long-term, satellite transmitting | Splash 10-292 long-term, satellite transmitting | Mk10 long-term, satellite transmitting | DTAG short-term, archival | 4 Mk10 long-term, satellite transmitting, 2 Mk9 suction-cup TDR short-term, archival |
| *sample size* | | | | | |
| no. of individuals | 10 | 8 | 6 | 7 | 6 |
| no. of deep dives | 1408 | 1142 | n.a. | 28 | 332 |
| location | Cape Hatteras, NC | southern California | Bahamas | Ligurian Sea, Italy | Hawaii |
| *deep dives* | | | | | |
| average depth | 1456 (median) 1492 (mean) | 1401 (group mean) | 1120 (median) | 1070 (mean) | 1284 (grand mean) |
| average duration | 58.9 (median) 58.7 (mean) | 67.4 (group mean) | 65 (median) | 58.0 (mean) | 61.3 (grand mean) |
| maximum depth | 2800 | 2992 | n.a. | 1888 | 2800 |
| maximum duration | 114.7 | 137.5 | 90.14 | 84.9 | 117 |
| IDDI | 74.6 (median) 76.6 (mean) | 102.3 (group mean) | 68 (median) | 63.4 (mean) | n.a. |
| dive rate (dives h$^{-1}$) | 0.43 | 0.30 | n.a. | n.a. | 0.44 |

(Continued.)

**Table 8.** (*Continued.*)

| | this study | Schorr et al. [3] | Joyce et al. [5] | Tyack et al. [4] | Baird [42] |
|---|---|---|---|---|---|
| number/day | ~10 | 7 | 10–11 | 11–12 | 10 |
| *shallow dives* | | | | | |
| average depth | 280 (median) 282 (mean) | 275 (group mean) | n.a. | 221 (mean) | n.a. |
| *surface* | | | | | |
| average duration | 2.2 (median) 4.7 (mean) | 1.87 (group mean) | n.a. | 7.9 (mean) | n.a. |
| maximum duration | 310 | 205.5 | n.a. | 85.6 | n.a. |

challenges to the function of satellite-linked tags employed in such studies. We deployed 11 SPLASH10-292 tags between 2014 and 2016, and two had catastrophic pressure sensor failures. One tag failed nearly immediately and the second failed after a month of deployment. It is likely that Cuvier's beaked whales are at least occasionally diving to depths which tags cannot withstand, causing them to fail. Tags with modifications to increase the tag housing pressure rating, and with pressure sensors capable of recording depths to 4000 m or more would reduce the risk of losing valuable data from these animals and verify the maximum depth to which they are capable of diving.

In addition, the short surface durations of these beaked whales provide only very brief opportunities to transmit data to orbiting satellites, resulting in frequent data gaps and incomplete tag records. In areas where animals occur close to land, shore-based receivers can improve data retrieval [50]. Changes to the protocols used to record and transmit data can reduce the number of data gaps, but at the expense of other aspects of the dive record [51]. Sensor errors and data transmission gaps have not been thoroughly addressed in the previous literature on beaked whales (but see [3,5,50]), and we argue that acknowledgement of the limitations of these tags will improve future study designs. Our analysis of the errors associated with these tags has indicated several areas where satellite tags could be improved. Given the extraordinary diving capabilities of these animals, there is a need to develop tags with sensors that are calibrated and rated to survive the extreme depths that these animals are capable of diving. To avoid losing information to data gaps from messages not fully transmitting at the surface, there is also a need for the tags to be able to contact a satellite and push their messages through more quickly to account for the short surface durations of these animals. Developing tags with these capabilities will greatly improve our knowledge of the world's deepest mammalian divers.

## 5. Conclusion

Cuvier's beaked whales off Cape Hatteras, North Carolina, like those tagged in other parts of the world, are extraordinary divers, but do not compensate for their very long submergences with long surface recovery periods. They dive nearly continuously and dive to foraging depths consistently both day and night. Long-term datasets, such as those presented here, are needed to examine variation in diving behaviour and to quantify other behavioural states such as long duration surface intervals. Finally, there is a need for beaked whale-specific tags that are calibrated to the remarkable depths at which these animals routinely dive.

Ethics. All research activities were carried out under National Marine Fisheries Service Scientific Research Permits nos. 17086 and 15330 issued to Robin Baird, and NOAA General Authorization 16185, issued to Andrew Read, in accordance with the relevant guidelines and regulations on the ethical use of animals as experimental subjects. The research approach was approved by the Institutional Animal Use and Care Committee (IACUC) of Cascadia Research Collective.
Data accessibility. Dive behaviour data are available from the Dryad Digital Repository: http://dx.doi.org/10.5061/dryad.6jv6vn8 [52]. Location data are publicly available on Movebank, study name 'Cuvier's beaked whale CRC NW Atlantic' (https://www.movebank.org/).
Authors' contributions. J.M.S. carried out the data analysis, participated in processing the data, contributed to analytical methods design and wrote the manuscript. N.J.Q. and W.R.C. participated in data processing and data analysis and contributed to analytical methods design. R.W.B. participated in designing the project and secured funding. D.L.W. participated in fieldwork and tagging. H.J.F. participated in fieldwork and in data processing. Z.T.S. participated in fieldwork and captained the research vessel. D.M.W. participated in fieldwork and processed photo-identification data. J.T.B. provided support and funding. A.J.R. participated in designing the project and secured funding. All authors contributed to editing and preparation of the final manuscript.
Competing interests. We declare we have no competing interests.
Funding. This project is funded by the US Fleet Forces Command.
Acknowledgements. We thank the numerous field teams that conducted tagging, Matthew Rutishauser at Wildlife Computers for tag technical support, and Brandon Southall and Doug Nowacek for their advice and assistance. We also thank the two anonymous reviewers whose comments greatly improved the manuscript.

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
