## [Reviewer comments · Royal Society Open Science]

Review History

RSOS-181728.R0 (Original submission)

Review form: Reviewer 1 (Diane Claridge)

Is the manuscript scientifically sound in its present form?

Yes

Are the interpretations and conclusions justified by the results?

Yes

Is the language acceptable?

Yes

Is it clear how to access all supporting data?

Yes

Do you have any ethical concerns with this paper?

No

Have you any concerns about statistical analyses in this paper?

No

Recommendation?

Accept with minor revision (please list in comments)

Comments to the Author(s)

This paper is a valuable contribution to our understanding of the extreme diving behavior in this species and is particularly relevant given the potential disturbance from human activities off Cape Hatteras.

Please see my attached file for detailed comments (Appendix A).

I found it annoying that the line numbers on each page did not align properly with the actual lines of text and suggest correcting this before submitting for further review.

Review form: Reviewer 2 (Gregory Schorr)**Is the manuscript scientifically sound in its present form?**

Yes

Are the interpretations and conclusions justified by the results?

Yes

Is the language acceptable?

Yes

Is it clear how to access all supporting data?

Yes

Do you have any ethical concerns with this paper?

No

Have you any concerns about statistical analyses in this paper?

No

Recommendation?

Accept with minor revision (please list in comments)

Comments to the Author(s)

This paper clearly and concisely summarizes the diving behavior of Cuvier's beaked whales tagged off Cape Hatteras. The methods and results are easy to follow, and the discussion clearly follows from the data presented. There are a few places that could benefit from clarification or rewording, and a few places where I believe references need to be added, but these are all minor changes.

Page 2, line 43: The Wildlife Computers LIMPET tag used in this study is now officially referred to by the manufacturer as the "SPLASH10-292" (See <https://wildlifecomputers.com/wp-content/uploads/mds/LIMPET-Tag-Suite-Specifications.pdf>). Suggest fixing throughout.

Page 2, line 47. Stating tags are 'designed to operate up to 1000 or 2000m' suggests that they

won't work beyond that point. Suggest clarifying that tags are calibrated to X meters with the corresponding manufacturer stated resolution, and clarify a pressure transducer that is designed not to exceed 3,000m.

Page 3 line 10: Argos is not an acronym, only the first letter should be capitalized. Fix throughout.

Page 3, line 44: You should reference testing these transducers to 3000m, e.g. Wildlife Computers or Schorr et al. (2014). This could help with the previous comment about tags functioning beyond their 'rated' depth.

Page 3, Line 52: The results (e.g. Page 5, line 46 and figure 4) suggest substantial variability between individual, with the range of individual median dive depths spanning almost 600m. In the discussion, individual variability is also highlighted. Based on this, why did you choose to pool your data in order to determine the deep vs shallow dive cutoff? The cutoff applied to one individual with a more limited sample could be substantially biased if it was influenced by a large number of data points from another animal behaving very differently. The scatter plot in figure 2 does not easily allow a reader to assess the degree to which individual data might be influencing pooled metrics, given the overall number of data points displayed. A hard cutoff doesn't deal well with the tails in the distribution of deep and shallow dives per individual (Figures 4 & 5). While there is likely only a fraction of dives that may be classified differently, given the emphasis on individual variability in the discussion, the authors should better qualify why/how pooling data for this important cutoff was chosen.

Page 5, line 60. Having referenced the term 'extreme' to previously describe beaked whale behavior, I understand the point- particularly with respect to other species. However, stating that their diving is both extreme and habitual in the same sentence seems a contradiction of terms. At this stage in our understanding of beaked whale diving, the data convincingly suggest that diving to depths greater than 1500m is routine for this species, so I would suggest moving away from representing 'habitual' behavior of beaked whales as extreme behavior.

Page 6, Line 17-24. This section is a bit hard to follow and understand. Overall, the paragraph reads as if focused on deep dive interval (IDDI) as a function of recovery time. The third sentence states that pilot whales average 8-12 minutes between foraging dives (i.e. a short IDDI) but then states that pilot whales spend long periods near the surface between foraging dives. That time should be included in the IDDI, so if that is 8-12 minutes, pilot whales would appear to spend (proportionately) much more time on deep foraging dives, not "a comparable amount over an entire diel cycle". Ultimately, this section seems to confound IDDI, foraging rates, surface intervals (as defined by tags in this study), and time near the surface (as referenced for other species) across species, and could stand to be rewritten to clarify the underlying point.

Page 6, line 60 to page 7 line 6. This section should reference Schorr et al. (2014), Tyack et al. (2015) and Schorr et al. (2015), all of which explore this point in detail for Cuvier's beaked whales. References not yet cited are:

Tyack PL, Calambokidis J, Friedlaender AS, Goldbogen JA (2015) Formal Comment on Schorr GS, Falcone EA, Moretti DJ, Andrews RD (2014) First Long-Term Behavioral Records from Cuvier's Beaked Whales (*Ziphius cavirostris*) Reveal Record-Breaking Dives (BL Southall, Ed.). PLoS ONE:1-4

Schorr GS, Falcone EA, Moretti DJ, Andrews RD (2015) Rebuttal to the Formal Comment on Schorr et al. (2014) submitted by Tyack et al. (2015) (A Fahlman, Ed.). PLOS ONE 10:e0142437

Page 7 line 56-58. Suggest including the above references (Tyack et al. 2015 and Schorr et al. 2015) here, as well, given they dealt directly with the issue of short versus long datasets. Also, suggest

changing the word from ‘Our study INDICATES...’ to ‘Our study CONFIRMS...’ since you reference a previous study having described this.

Page 8, line 20-23: In addition to this unpublished manuscript, it seems like you should also reference a paper demonstrating how data gaps can be reduced by utilizing receivers other than just those on orbiting satellites. Suggest adding after “...incomplete tag records...” ‘unless shore-based receivers are utilized (Jeanniard-du-Dot et al., 2017)’, or similar. Also, if the journal has issues with references that can’t be accessed in any way at the time of publication, in addition to the unpublished manuscript you might also reference a gray literature article dealing specifically with this issue.

Jeanniard-du-Dot T, Holland K, Schorr GS, Vo D (2017) Motes enhance data recovery from satellite-relayed biologgers and can facilitate collaborative research into marine habitat utilisation. *Animal Biotelemetry* 5:1-15

Schorr GS, Rone BK, Falcone Erin A (2017) Integrated measurement of Naval sonar operations and precise cetacean locations: Integration of Fastloc GPS into a LIMPET tag. Final report for Task C: Contrac No. N66604-14-C-2438. Availabe at www.marecotel.org

Page 8 line 24: I agree that sensor issues have not always been addressed in detail in previous peer-reviewed literature, and I commend the authors for more fully discussing this issue and how it was dealt with. However in regards to data gaps and other limitations of the tags not having been acknowledged, several studies on beaked whales using LIMPET tags referenced in your paper have acknowledged/addressed at least some of the limitations of these tags (e.g. the presence and influence of data gaps, the limitations of summarized data, etc), at least to some degree (see Schorr et al., 2014 & 2015, Joyce et al. 2016, Jeanniard-du-Dot et al., 2017). This statement should identify specific data issues that have not been adequately addressed, and/or these references should be added.

Page 8 line 27: Clarify how your data identifies the need for tags to transmit data “more quickly”, since this doesn’t seem clear from the above. Is this need based on Argos message length or size (e.g. faster transmission would allow for transmission of longer or larger messages)? Also, stating there is a need for tags capable of “handling the extraordinary diving capabilities” suggests that the tags you and previous authors are using are not. As this is the final sentence of the discussion, it should be more clearly supported by the data discussed above.

Page 8 line 42. Following the previous comment, again what exactly is meant by ‘...develop tags capable of accurately measuring behavior at the extreme depths at which these animals forage’? Are you referring only to the need for a LIMPET tag with a deeper rated pressure sensor? Suggest re-wording this sentence since you describe all of the deep dives in your dataset as extreme, so this could be read to suggest that the tag used here (or any other tag currently available) is not capable of ‘accurately’ measuring behavior.

Page 22, line 30. You state the 800 m cutoff has been used “previously” to segregate deep and shallow dives classes, but this does not appear to be referenced elsewhere in the manuscript. See also previous comment about using this cutoff limit. Please add a reference or clarify.

Decision letter (RSOS-181728.R0)

17-Dec-2018

Dear Ms Shearer

On behalf of the Editors, I am pleased to inform you that your Manuscript RSOS-181728 entitled "DIVING BEHAVIOR OF CUVIER'S BEAKED WHALES (*Ziphius cavirostris*) OFF CAPE HATTERAS, NORTH CAROLINA" has been accepted for publication in Royal Society Open Science subject to minor revision in accordance with the referee suggestions. Please find the referees' comments at the end of this email.

The reviewers and handling editors have recommended publication, but also suggest some minor revisions to your manuscript. Therefore, I invite you to respond to the comments and revise your manuscript.

- Ethics statement

- Data accessibility

If you wish to submit your supporting data or code to Dryad (<http://datadryad.org/>), or modify your current submission to dryad, please use the following link:
<http://datadryad.org/submit?journalID=RSOS&manu=RSOS-181728>

- Competing interests

- Authors' contributions

- Acknowledgements

- Funding statement

Because the schedule for publication is very tight, it is a condition of publication that you submit the revised version of your manuscript before 26-Dec-2018. Please note that the revision deadline will expire at 00.00am on this date. If you do not think you will be able to meet this date please let me know immediately.

- 1) A text file of the manuscript (tex, txt, rtf, docx or doc), references, tables (including captions) and figure captions. Do not upload a PDF as your "Main Document";
- 2) A separate electronic file of each figure (EPS or print-quality PDF preferred (either format should be produced directly from original creation package), or original software format);
- 3) Included a 100 word media summary of your paper when requested at submission. Please ensure you have entered correct contact details (email, institution and telephone) in your user account;
- 4) Included the raw data to support the claims made in your paper. You can either include your data as electronic supplementary material or upload to a repository and include the relevant doi within your manuscript. Make sure it is clear in your data accessibility statement how the data can be accessed;

5) All supplementary materials accompanying an accepted article will be treated as in their final form. Note that the Royal Society will neither edit nor typeset supplementary material and it will be hosted as provided. Please ensure that the supplementary material includes the paper details where possible (authors, article title, journal name).

on behalf of Dr Asha de Vos (Associate Editor) and Kevin Padian (Subject Editor)
openscience@royalsociety.org

Associate Editor Comments to Author (Dr Asha de Vos):

Associate Editor: 1

Comments to the Author:

Well done on an interesting and well prepared manuscript. We hope the minor revisions suggested by our reviewers will help the manuscript further and look forward to its publication!

Reviewer comments to Author:

Reviewer: 1

Comments to the Author(s)

This paper is a valuable contribution to our understanding of the extreme diving behavior in this species and is particularly relevant given the potential disturbance from human activities off Cape Hatteras.

Please see my attached file for detailed comments.

I found it annoying that the line numbers on each page did not align properly with the actual lines of text and suggest correcting this before submitting for further review.

Reviewer: 2

Comments to the Author(s)

This paper clearly and concisely summarizes the diving behavior of Cuvier's beaked whales tagged off Cape Hatteras. The methods and results are easy to follow, and the discussion clearly follows from the data presented. There are a few places that could benefit from clarification or rewording, and a few places where I believe references need to be added, but these are all minor changes.

Page 2, line 43: The Wildlife Computers LIMPET tag used in this study is now officially referred to by the manufacturer as the "SPLASH10-292" (See <https://wildlifecomputers.com/wp-content/uploads/mds/LIMPET-Tag-Suite-Specifications.pdf>). Suggest fixing throughout.

Page 2, line 47. Stating tags are 'designed to operate up to 1000 or 2000m' suggests that they won't work beyond that point. Suggest clarifying that tags are calibrated to X meters with the corresponding manufacturer stated resolution, and clarify a pressure transducer that is designed not to exceed 3,000m.

Page 3 line 10: Argos is not an acronym, only the first letter should be capitalized. Fix throughout.

Page 3, line 44: You should reference testing these transducers to 3000m, e.g. Wildlife Computers or Schorr et al. (2014). This could help with the previous comment about tags functioning beyond their 'rated' depth.

Page 3, Line 52: The results (e.g. Page 5, line 46 and figure 4) suggest substantial variability between individual, with the range of individual median dive depths spanning almost 600m. In the discussion, individual variability is also highlighted. Based on this, why did you choose to pool your data in order to determine the deep vs shallow dive cutoff? The cutoff applied to one individual with a more limited sample could be substantially biased if it was influenced by a large number of data points from another animal behaving very differently. The scatter plot in figure 2 does not easily allow a reader to assess the degree to which individual data might be influencing pooled metrics, given the overall number of data points displayed. A hard cutoff doesn't deal well with the tails in the distribution of deep and shallow dives per individual (Figures 4 & 5). While there is likely only a fraction of dives that may be classified differently, given the emphasis on individual variability in the discussion, the authors should better qualify why/how pooling data for this important cutoff was chosen.

Page 5, line 60. Having referenced the term 'extreme' to previously describe beaked whale behavior, I understand the point- particularly with respect to other species. However, stating that their diving is both extreme and habitual in the same sentence seems a contradiction of terms. At this stage in our understanding of beaked whale diving, the data convincingly suggest that diving to depths greater than 1500m is routine for this species, so I would suggest moving away from representing 'habitual' behavior of beaked whales as extreme behavior.

Page 6, Line 17-24. This section is a bit hard to follow and understand. Overall, the paragraph reads as if focused on deep dive interval (IDDI) as a function of recovery time. The third sentence states that pilot whales average 8-12 minutes between foraging dives (i.e. a short IDDI) but then states that pilot whales spend long periods near the surface between foraging dives. That time should be included in the IDDI, so if that is 8-12 minutes, pilot whales would appear to spend (proportionately) much more time on deep foraging dives, not "a comparable amount over an

entire diel cycle". Ultimately, this section seems to confound IDDI, foraging rates, surface intervals (as defined by tags in this study), and time near the surface (as referenced for other species) across species, and could stand to be rewritten to clarify the underlying point.

Page 6, line 60 to page 7 line 6. This section should reference Schorr et al. (2014), Tyack et al. (2015) and Schorr et al. (2015), all of which explore this point in detail for Cuvier's beaked whales. References not yet cited are:

Tyack PL, Calambokidis J, Friedlaender AS, Goldbogen JA (2015) Formal Comment on Schorr GS, Falcone EA, Moretti DJ, Andrews RD (2014) First Long-Term Behavioral Records from Cuvier's Beaked Whales (*Ziphius cavirostris*) Reveal Record-Breaking Dives (BL Southall, Ed.). PLoS ONE:1-4

Schorr GS, Falcone EA, Moretti DJ, Andrews RD (2015) Rebuttal to the Formal Comment on Schorr et al. (2014) submitted by Tyack et al. (2015) (A Fahlman, Ed.). PLOS ONE 10:e0142437

Page 7 line 56-58. Suggest including the above references (Tyack et al. 2015 and Schorr et al. 2015) here, as well, given they dealt directly with the issue of short versus long datasets. Also, suggest changing the word from 'Our study INDICATES...' to 'Our study CONFIRMS...' since you reference a previous study having described this.

Page 8, line 20-23: In addition to this unpublished manuscript, it seems like you should also reference a paper demonstrating how data gaps can be reduced by utilizing receivers other than just those on orbiting satellites. Suggest adding after "...incomplete tag records..." 'unless shore-based receivers are utilized (Jeanniard-du-Dot et al., 2017)', or similar. Also, if the journal has issues with references that can't be accessed in any way at the time of publication, in addition to the unpublished manuscript you might also reference a gray literature article dealing specifically with this issue.

Jeanniard-du-Dot T, Holland K, Schorr GS, Vo D (2017) Motes enhance data recovery from satellite-relayed biologgers and can facilitate collaborative research into marine habitat utilisation. *Animal Biotelemetry* 5:1-15

Schorr GS, Rone BK, Falcone Erin A (2017) Integrated measurement of Naval sonar operations and precise cetacean locations: Integration of Fastloc GPS into a LIMPET tag. Final report for Task C: Contrac No. N66604-14-C-2438. Available at www.marecotel.org

Page 8 line 24: I agree that sensor issues have not always been addressed in detail in previous peer-reviewed literature, and I commend the authors for more fully discussing this issue and how it was dealt with. However in regards to data gaps and other limitations of the tags not having been acknowledged, several studies on beaked whales using LIMPET tags referenced in your paper have acknowledged/addressed at least some of the limitations of these tags (e.g. the presence and influence of data gaps, the limitations of summarized data, etc), at least to some degree (see Schorr et al., 2014 & 2015, Joyce et al. 2016, Jeanniard-du-Dot et al., 2017). This statement should identify specific data issues that have not been adequately addressed, and/or these references should be added.

Page 8 line 27: Clarify how your data identifies the need for tags to transmit data "more quickly", since this doesn't seem clear from the above. Is this need based on Argos message length or size (e.g. faster transmission would allow for transmission of longer or larger messages)? Also, stating there is a need for tags capable of "handling the extraordinary diving capabilities" suggests that the tags you and previous authors are using are not. As this is the final sentence of the discussion, it should be more clearly supported by the data discussed above.

Page 8 line 42. Following the previous comment, again what exactly is meant by ‘...develop tags capable of accurately measuring behavior at the extreme depths at which these animals forage’? Are you referring only to the need for a LIMPET tag with a deeper rated pressure sensor? Suggest re-wording this sentence since you describe all of the deep dives in your dataset as extreme, so this could be read to suggest that the tag used here (or any other tag currently available) is not capable of ‘accurately’ measuring behavior.

Page 22, line 30. You state the 800 m cutoff has been used “previously” to segregate deep and shallow dives classes, but this does not appear to be referenced elsewhere in the manuscript. See also previous comment about using this cutoff limit. Please add a reference or clarify.

Author's Response to Decision Letter for (RSOS-181728.R0)

See Appendix B.

Decision letter (RSOS-181728.R1)

09-Jan-2019

Dear Ms Shearer,

I am pleased to inform you that your manuscript entitled "DIVING BEHAVIOR OF CUVIER'S BEAKED WHALES (*Ziphius cavirostris*) OFF CAPE HATTERAS, NORTH CAROLINA" is now accepted for publication in Royal Society Open Science.

on behalf of Dr Asha de Vos (Associate Editor) and Kevin Padian (Subject Editor)
openscience@royalsociety.org

Appendix A

Ziphius Diving Behavior Cape Hatteras

General Overview

This paper contributes valuable information on the diving behavior of a species that is vulnerable to disturbance in an area of high human use.

Comments:

Summary

Page 1 Line 34: Given the difficulty in sexing Zc at sea, how confident are you that all animals are adults?

I suggest adding a sentence about the value of these data for a species vulnerable to noise disturbance in an area with regular military activity and oil and gas exploration.

Introduction

Page 1 Line 3: Should scientific names be given for all species?

Page 2 Line 8: Add clarification that the deepest and longest duration dive that Schorr et al 2014 report could have been by a whale exposed to navy sonar.

Were any of the tagged whales in this study subjected to CEEs or real sonar activities?

Page 2 Line 17: Is succeeding the correct word, not preceding?

Page 2 Lines 18-19: One case or reported in one study?

Starting at paragraph 2 (Line 15), I suggest focusing on what's known about the diving behavior just for Ziphius and not for all beaked whale species (where data exist). This would add clarity to the writing and eliminate awkward sentences, e.g., Page 2 Lines 26-29 could be stated more clearly if just providing background on Ziphius and not M densirostris as well.

Page 2 Line 31: Forney et al 2017 is not the correct citation for strong site fidelity of Ziphius at Cape Hatteras. That paper references Baird RW, Webster DL, Swaim Z, Foley HJ, Anderson DB, Read AJ (2016) Spatial use by odontocetes satellite tagged off Cape Hatteras, North Carolina, in 2015. Report prepared for US Fleet Forces Command, Norfolk, VA. The telemetry

data presented in Baird et al's report appears to be the same as what you are presenting here (your Fig 1).

Page 2 Line 33: Add a sentence here about the human uses of the study area (military exercises, oil and gas exploration, fishing, ship traffic) to demonstrate the need for understanding diving behavior for this species in this location as that is the data gap addressed in this study. You could reference Forney et al here.

Materials and Methods

Page 2 Lines 45-46: Add citation after tag configuration Andrews R, Pitman R, Ballance L (2008) Satellite tracking reveals distinct movement patterns for Type B and Type C killer whales in the southern Ross Sea, Antarctica. *Polar Biol* 31: 1461–1468. doi:10.1007/s00300-008-0487-z.

Page 2 Lines 45-46: “on” the dorsal fin, not “into”

Page 2 Lines 47-49: Can you explain what is meant by the tags were designed to operate to 1000 and 2000m depths?

Page 2 Line 49: What type of pneumatic rifle was used? Give manufacturer's info, etc.

Page 2 Lines 53-54: Re: photo-identification, add that you compared identifications with a catalogue of known individuals from the study area to aid in designating sex for individuals with prior sighting histories.

Page 2 Lines 54-55: Add ref McCANN, C. 1974. Body scarring on Cetacea-Odontocetes. *Scientific Report of the Whales Research Institute* 26:145-155.

Also, add that white pigmentation on the male's head helped with classification, as you later note.

Page 2 Line 58: You should add a cautionary statement about the difficulty in distinguishing adult females from subadult males.

Page 2 Lines 58-60: You need to clarify that you mean you were able to classify adult males more easily due to the white pigmentation on the head, not that the pigmentation predisposed them to tagging! Also, what behaviour are you referring to? Lagging behind the rest of the group? Surfacing more often?

Page 3 Lines 4-5: The Satellite Tag section could be named Satellite Tag Programming

Page 3 Line 7: How were the tags programmed and why 20 hrs per day?

Page 3 Lines 9-13: The gaps in tag data record that you encountered could be moved to Error Checks or Results sections. That way, you can also address how you dealt with these gaps.

Page 3 Line 15: Why 50 m?

Page 3 Lines 16-18: Explain why max depths are recorded as 2 values.

Page 3 Lines 26-27: e.g. instead of i.e.

Page 3 Lines 29-35: Tag programming for the time series data may be better placed in the section above.

Page 3 Line 46: Weren't all other dives were retained not just the subsequent dives?

Page 4 Lines 15-17: What manner are you referring to?

Page 4 Lines 33-35: I'd be cautious about stating adult female classifications and also about adult classifications for the individuals of unknown sex. Is it possible that one or more of the females could be sub-adult males and/or the unknown adults could be sub-adults? How many of the tagged whales were photo-identified? Were there any animals known previously?

Page 4 Lines 38-40: Move and combine this sentence to the above section (line 8) where you say why both medians and means are given, i.e., to show skewness and allow comparison to other studies.

Page 5 Line 2: You need to define the word "surface", i.e., any time <50m and not at the surface for the entire time, except when logging.

Page 5 Lines 4-5: At or near the surface.

Page 5 Lines 7-8: Do the time series data suggest these animals were logging during these extended bouts, especially the 310 min surfacing interval?

Page 5 Lines 33-35: You have already noted that there were more shallow dives during the day (line 32).

Page 5 Lines 42-47: Omit this section as it is relevant to the Discussion.

Discussion

Page 5 Line 53 – Page 6 line 33: This section on extreme diving and IDDI doesn't present anything new to past discussions or use the findings from this study to advance the discussion so although it is well-written, I suggest removing this section from the manuscript.

Page 6 Lines 44-46: How does the water depth vary between these locations and what max depths were possible given the whales' locations? Were the whales diving to the bottom in all cases?

Page 6 Lines 53-54: Give the areas where the parameters are available, namely California and Liguria.

Page 7 Lines 8-10: Note that to include more information on age and sex of tagged whales, getting ID photos and biopsy samples from tagged whales will need to be more of a priority.

Page 7 Lines 21-23: Future use of a goniometer for real-time tracking will help improve location resolution and aid in more accurate bathymetry.

Page 7 Lines 39-46: Are there sightings data for killer whales off Cape Hatteras that can be referenced here to help explain the diel difference found in this study?

Page 7 Line 48 – Page 8 Line 6: I found this section weak in content and poorly written, e.g., there aren't "many" prior studies using short-term tags. It could be rewritten more succinctly and needs additional references added. I think that the combination of behavioural data from short-term, high resolution tags with long-term tags is ideal. Each has its own strengths. Also, it is probably clearer to use suction-cup/Dtags and LIMPET tags rather than short and long term because some of the LIMPET deployments are actually short-term.

Page 8 Line 2: I think you mean that not all individuals were fully sampled. Also adding the use of a goniometer will aid in re-sighting the tagged whale to increase opportunities for photo-id and biopsy to confirm sex. Also, I'd like to see some discussion about TagZc051. Why do you think it behaved differently?

Page 8 Lines 14-16: I would reword to say it is likely that Cuvier's are at least occasionally diving...

Table 3. Add max for the shallow dive depths and durations.

Table 7. Define "Surface" again here in the legend.

Appendix B

21 December 2018

Dear Dr. Sanders

Enclosed and uploaded please find our revised manuscript entitled "Diving behavior of Cuvier's beaked whales (*Ziphius cavirostris*) off Cape Hatteras, North Carolina. We would like to express our sincere appreciation to you, the Associate Editor, and the two Reviewers. The comments have greatly improved our manuscript.

Below we respond to each Reviewer's comments in turn (our responses are in blue) and describe how we have modified the manuscript in response to their suggestions. We have also uploaded a version of the manuscript with tracked changes to show the edits we made.

Please let us know if there is any other information you require to consider this revised manuscript.

Sincerely,

Jeanne Shearer
Andrew Read

Reviewer 1:

General Overview

This paper contributes valuable information on the diving behavior of a species that is vulnerable to disturbance in an area of high human use.

We greatly appreciate this reviewer for their helpful and thorough comments which have significantly improved our manuscript. We have answered specific comments and questions below.

Comments:

Page 1 Line 34: Given the difficulty in sexing Zc at sea, how confident are you that all animals are adults?

We thank the reviewer for this point and agree with the difficulty of sexing Zc. We did not determine gender or age class at sea, but rather made these determinations in the laboratory through photo-identification and review of tagging videos. We are very confident of the gender of six adult males and one of the adult females (ZcTag047), which was seen in previous years with a dependent calf. Researchers from two laboratories independently assessed the images and determined gender and age classes. Given the difficulties in determining age class of unmarked animals,

we have updated Table 2 to state 'likely' adult females for those animals that were not seen with a calf and deleted the 'adult whale' qualifier from this sentence in the abstract, per the reviewer's suggestion.

I suggest adding a sentence about the value of these data for a species vulnerable to noise disturbance in an area with regular military activity and oil and gas exploration. We have made this change on Page 2 Line 33(see below) but have not added anything to the Abstract as we are already at the word limit.

Page 1 Line 3: Should scientific names be given for all species?
We have made this change.

Page 2 Line 8: Add clarification that the deepest and longest duration dive that Schorr et al 2014 report could have been by a whale exposed to navy sonar. We have added the following clarification: "...southern California, although it is possible that these animals were exposed to Navy sonar during those deployments." and have cited the formal comment to the Schorr et al. 2014 paper as well as the rebuttal to that comment.

Were any of the tagged whales in this study subjected to CEEs or real sonar activities?
None of the whales in this study were subject to CEEs and to the best of our knowledge, were not exposed to real sonar activities.

Page 2 Line 17: Is succeeding the correct word, not preceding?
For clarity, we have edited that sentence to use the word 'following' rather than succeeding, as well as making other edits to the sentence based on reviewer feedback (see comments below).

Page 2 Lines 18-19: One case or reported in one study?
This was no longer included per the suggestion to focus on Cuvier's beaked whales (see comment below).

Starting at paragraph 2 (Line 15), I suggest focusing on what's known about the diving behavior just for Ziphius and not for all beaked whale species (where data exist). This would add clarity to the writing and eliminate awkward sentences, e.g., Page 2 Lines 26-29 could be stated more clearly if just providing background on Ziphius and not M densirostris as well.

We agree that this would add clarity to this paragraph. We have edited that paragraph to include only Cuvier's beaked whales and have adjusted the references accordingly.

Page 2 Line 31: Forney et al 2017 is not the correct citation for strong site fidelity of Ziphius at Cape Hatteras. That paper references Baird RW, Webster DL, Swaim Z, Foley HJ, Anderson DB, Read AJ (2016) Spatial use by odontocetes satellite tagged off Cape Hatteras, North Carolina, in 2015. Report prepared for US Fleet Forces Command, Norfolk, VA. The telemetry data presented in Baird et al's report appears to be the same as what you are presenting here (your Fig 1).

We have updated the references to remove the Forney et al. 2017 paper and added the original papers Baird et al. 2015, 2016, 2017, which utilize the satellite tags reported on here as well as additional location only tags not included in the current study.

Page 2 Line 33: Add a sentence here about the human uses of the study area (military exercises, oil and gas exploration, fishing, ship traffic) to demonstrate the need for understanding diving behavior for this species in this location as that is the data gap addressed in this study. You could reference Forney et al here.
We have added the following sentence: "These data address a gap in knowledge for this species occurring in an area with multiple human uses, from occasional military exercises to extensive fishing and shipping traffic and the recent potential for offshore seismic surveys (Forney et al. 2017)."

Page 2 Lines 45-46: Add citation after tag configuration Andrews R, Pitman R, Ballance L (2008) Satellite tracking reveals distinct movement patterns for Type B and Type C killer whales in the southern Ross Sea, Antarctica. Polar Biol 31: 1461–1468. doi:10.1007/s00300-008-0487-z.

We have added this citation.

Page 2 Lines 45-46: "on" the dorsal fin, not "into"
We have made this edit.

Page 2 Lines 47-49: Can you explain what is meant by the tags were designed to operate to 1000 and 2000m depths?

This was also addressed by Reviewer 2. The text now reads: “Nine tags were in an extended depth configuration (calibrated to 2000 m with a 1 m resolution, with a pressure sensor designed not to exceed 3000 m). Two additional tags (ZcTag046 and ZcTag047) were in a standard depth configuration (calibrated to 1000 m with a 0.5 m resolution, with a pressure sensor designed to survive depths of 2000 m).”

Page 2 Line 49: What type of pneumatic rifle was used? Give manufacturer’s info, etc.

We have added the manufacturer’s info (DAN-INJECT JM 25, DanWild LLC, Austin, Texas).

Page 2 Lines 53-54: Re: photo-identification, add that you compared identifications with a catalogue of known individuals from the study area to aid in designating sex for individuals with prior sighting histories.

We have added the following sentence: “Photos were compared with a catalogue of known individuals from the study area to aid in designating sex and age classes for individuals with a prior or subsequent sighting record.”

Page 2 Lines 54-55: Add ref MCCANN, C. 1974. Body scarring on Cetacea-Odontocetes. Scientific Report of the Whales Research Institute 26:145-155.

We have added this citation.

Also, add that white pigmentation on the male’s head helped with classification, as you later note.

We have added this clarification.

Page 2 Line 58: You should add a cautionary statement about the difficulty in distinguishing adult females from subadult males.

We agree with the reviewer and have added the following sentence: “Without the presence of a calf, it is difficult to distinguish adult females from subadult males. All classifications were made independently by two laboratories and sighting histories and videos of tagging attempts for some of the animals increase our confidence in our classifications.”

Page 2 Lines 58-60: You need to clarify that you mean you were able to classify adult males more easily due to the white pigmentation on the head, not that the pigmentation predisposed them to tagging! Also, what behavior are you referring to? Lagging behind the rest of the group? Surfacing more often?

We agree with the reviewer that this sentence was ambiguous and have changed it to read: "Our sample of tagged individuals is biased toward males because the distinctive white pigmentation makes them more visible to observers and easier to track during tagging attempts. The white pigmentation and linear scarring in males increases our confidence in their classification." We have removed the statement about behavior as we have only anecdotal evidence for this claim.

Page 3 Lines 4-5: The Satellite Tag section could be named Satellite Tag Programming

We have made this edit.

Page 3 Line 7: How were the tags programmed and why 20 hrs per day?

Tags were programmed using Wildlife Computers Mk10Host program. They were set to transmit for 20 hours a day based on an analysis of satellite overpasses. Periods of low quality or no satellite passes were turned off to conserve battery.

Page 3 Lines 9-13: The gaps in tag data record that you encountered could be moved to

Error Checks or Results sections. That way, you can also address how you dealt with these gaps.

We agree with this suggestion and have moved the sentences on gaps to the end of the first paragraph under 'error checks' and have added the following sentence: "We omitted these gaps from all calculated statistics (e.g. inter-deep dive intervals were not calculated when a gap occurred between deep dives, and surface duration statistics were not calculated when a gap occurred immediately before or after a dive)."

Page 3 Line 15: Why 50 m?

Time at the surface was defined as time between dives exceeding 50 meters to avoid generating excessive numbers of shallow inter-ventilation submergences, which would increase the number of messages to transmit. This is also the definition used by Baird et al. 2008 and Schorr et al. 2014.

Page 3 Lines 16-18: Explain why max depths are recorded as 2 values.

Wildlife Computers records depth in the behavior log as a maximum and minimum value for the depth of the dive and they state that the true value of the dive depth is between these two values.

Page 3 Lines 26-27: e.g. instead of i.e.

We have made this edit.

Page 3 Lines 29-35: Tag programming for the time series data may be better placed in the section above.

We agree and have made this change.

Page 3 Line 46: Weren't all other dives were retained not just the subsequent dives?
We have changed the line to read "... we retained all other dives in our analysis."

Page 4 Lines 15-17: What manner are you referring to?

We were referring to the fact that the behavior log only records the depth and duration of the dive. It does not record a time series of the dive (except for occasional duty-cycled time series measurements used to verify summarized data) and thus we have no means of calculating the percent of time spent in each depth layer on the dive.

Page 4 Lines 33-35: I'd be cautious about stating adult female classifications and also about adult classifications for the individuals of unknown sex. Is it possible that one or more of the females could be sub-adult males and/or the unknown adults could be sub-adults? How many of the tagged whales were photo-identified? Were there any animals known previously?

We agree that it is possible that some of our animals of unknown sex could be sub-adults and we have updated Table 2 to reflect this possibility. We have changed these lines in the methods to state: "Most (six) of the tagged whales were adult males, one was an adult female, two were likely adult females, and two were likely adult males."

Three of 11 animals had prior sighting histories (ZcTag030, ZcTag040 and ZcTag047) while 4 of 11 have been resighted after tagging (ZcTag029, ZcTag040, ZcTag046, ZcTag047). ZcTag047 (adult female) was seen two years prior to tagging with a dependent calf. ZcTag029 (likely adult female) was seen 4 years following tagging; its size, lack of scarring and clear head shot showing no erupted teeth in this sighting increase our confidence in classifying this animal as an adult female.

Page 4 Lines 38-40: Move and combine this sentence to the above section (line 8) where you say why both medians and means are given, i.e., to show skewness and allow comparison to other studies.
Thank you, we have made this edit.

Page 5 Line 2: You need to define the word "surface", i.e., any time <50m and not at the surface for the entire time, except when logging.
We agree with the reviewer and have changed the heading from "Surface Behavior" to "Surface (<50 m depth) Behavior".

Page 5 Lines 4-5: At or near the surface.

Thank you, we have made this edit.

Page 5 Lines 7-8: Do the time series data suggest these animals were logging during these

extended bouts, especially the 310 min surfacing interval?

There are no time series data for the 310 min surfacing interval. For periods in which long surface intervals overlap with time series data, animals do not appear to be logging but instead perform dives of 10-40 meters.

Page 5 Lines 33-35: You have already noted that there were more shallow dives during the day (line 32).

We have edited this sentence to remove that statement.

Page 5 Lines 42-47: Omit this section as it is relevant to the Discussion.

We agree with the reviewer and have removed this section.

Page 5 Line 53 – Page 6 line 33: This section on extreme diving and IDDI doesn't present

anything new to past discussions or use the findings from this study to advance the discussion so although it is well-written, I suggest removing this section from the manuscript.

We have removed most of this section from the manuscript. We have retained the lines at the beginning summarizing the findings and a few from the end of the section raising new research questions for these animals.

Page 6 Lines 44-46: How does the water depth vary between these locations and what max

depths were possible given the whales' locations? Were the whales diving to the bottom in all cases?

We agree with the reviewer that this information would be very interesting and would certainly add useful context. Unfortunately, given the complex topography of the area and the large error radius of the position data from Service Argos, it is impossible to know whether they are diving to the bottom. This is information we would love to have and hope that in the future it will be available.

Page 6 Lines 53-54: Give the areas where the parameters are available, namely California and Liguria.

Thank you, we have made this addition.

Page 7 Lines 8-10: Note that to include more information on age and sex of tagged whales,

getting ID photos and biopsy samples from tagged whales will need to be more of a priority.

We agree with the reviewer on this point and have edited this sentence to read: "Increasing sample size and including information on the sex and age of the tagged animal (through increasing effort to obtain biopsy samples and ID photos) would help determine if there are age or sex class differences in behavior in addition to population level differences."

Emphasis on collecting photos of both sides of all group members, as well as targeting tagged animals for biopsy sampling have been a priority in later field seasons in this area. Seven biopsies were collected in 2018 (2 of tagged animals), more than all prior field seasons combined.

Page 7 Lines 21-23: Future use of a goniometer for real-time tracking will help improve location resolution and aid in more accurate bathymetry. We agree. We have utilized a goniometer in later field seasons in this area.

Page 7 Lines 39-46: Are there sightings data for killer whales off Cape Hatteras that can be referenced here to help explain the diel difference found in this study? There are no published sightings but there is considerable local environmental knowledge (including video footage) of killer whale occurrence in this area, especially during winter.

Page 7 Line 48 – Page 8 Line 6: I found this section weak in content and poorly written, e.g., there aren't "many" prior studies using short-term tags. It could be rewritten more succinctly and needs additional references added. I think that the combination of behavioral data from short-term, high resolution tags with long-term tags is ideal. Each has its own strengths. Also, it is probably clearer to use suction-cup/Dtags and LIMPET tags rather than short and long term because some of the LIMPET deployments are actually short-term. We agree and have rewritten the paragraph and added references, and direct the reader to the revised manuscript for those edits.

Page 8 Line 2: I think you mean that not all individuals were fully sampled. Also adding the use of a goniometer will aid in re-sighting the tagged whale to increase opportunities for photo-id and biopsy to confirm sex.

We have changed the wording as suggested. In later field seasons we have been using a goniometer to resight tagged whales.

Also, I'd like to see some discussion about TagZc051. Why do you think it behaved differently?

We thank the reviewer for this observation and agree that it would be interesting to address the behavioral differences for ZcTag051. However, addressing individual level variation is beyond the scope of this manuscript. In fact, we are preparing a separate manuscript using data from these tags as well as additional deployments (in which tags were programmed in a different way making it difficult or impossible to include them in this analysis).

Page 8 Lines 14-16: I would reword to say it is likely that Cuvier's are at least occasionally diving...

Thank you, we have made this edit.

Table 3. Add max for the shallow dive depths and durations.

We have made this edit.

Table 7. Define "Surface" again here in the legend.

We have added "Deep dives are greater than 800 m, shallow dives are to depths of 50-800 m and surface intervals are between dives of 50 or more m." to the caption.

Reviewer 2:

Comments to the Author(s)

This paper clearly and concisely summarizes the diving behavior of Cuvier's beaked whales tagged off Cape Hatteras. The methods and results are easy to follow, and the discussion clearly follows from the data presented. There are a few places that could benefit from clarification or re-wording, and a few places where I believe references need to be added, but these are all minor changes.

We are grateful to this reviewer for her/his detailed and helpful comments. We have made edits in the text where rewording and clarification were requested, and we address specific comments and questions raised by this reviewer here.

Page 2, line 43: The Wildlife Computers LIMPET tag used in this study is now officially referred to by the manufacturer as the "SPLASH10-292"

(See [https://urldefense.proofpoint.com/v2/url?u=https-3A_wildlifecomputers.com_wp-2Dcontent_uploads_mds_LIMPET-2DTag-2DSuite-2DSpecifications.pdf&d=DwIFaQ&c=imBPVzF25OnBgGmVOlcsiEgHoG1i6YHLROsj_gZ4adc&r=dS5sf0uS8r7FKKyT7aWc4XR7avm6iO2s4SOM3VStqqs&m=NKOkTMLnRbmGICOoyn30C0MhWtMBZbn6qHpCuhcBaBk&s=4Bq9NWtCN1Z1SjQMDAQERL-q5nmP79T2I82B4Ffi5w&e](https://urldefense.proofpoint.com/v2/url?u=https-3A_wildlifecomputers.com_wp-2Dcontent_uploads_mds_LIMPET-2DTag-2DSuite-2DSpecifications.pdf&d=DwIFaQ&c=imBPVzF25OnBgGmVOlcsiEgHoG1i6YHLROsj_gZ4adc&r=dS5sf0uS8r7FKKyT7aWc4XR7avm6iO2s4SOM3VStqqs&m=NKOkTMLnRbmGICOoyn30C0MhWtMBZbn6qHpCuhcBaBk&s=4Bq9NWtCN1Z1SjQMDAQERL-q5nmP79T2I82B4Ffi5w&e=)). Suggest fixing throughout.

We have updated the text throughout the paper.

Page 2, line 47. Stating tags are 'designed to operate up to 1000 or 2000m' suggests that they won't work beyond that point. Suggest clarifying that tags are calibrated to X meters with the corresponding manufacturer stated resolution, and clarify a pressure transducer that is designed not to exceed 3,000m.

We agree with the reviewer. We have edited this to read: "Nine tags were in an extended depth configuration (calibrated to 2000 m with a 1 m resolution, with a pressure sensor designed not to exceed 3000 m). Two additional tags (ZcTag046 and ZcTag047) were in a standard depth configuration (calibrated to 1000 m with a 0.5 m resolution, with a pressure sensor designed to survive depths of 2000 m)."

Of the two tags that were in standard depth configuration, ZcTag047 failed and data from this tag was not used in the analysis. ZcTag046 did not dive beyond 2000 m, allowing us to retain these data in our analysis despite the shallower configuration.

Page 3 line 10: Argos is not an acronym, only the first letter should be capitalized. Fix throughout.

We have corrected this error.

Page 3, line 44: You should reference testing these transducers to 3000m, e.g. Wildlife Computers or Schorr et al. (2014). This could help with the previous comment about tags functioning beyond their 'rated' depth.

We have added the appropriate reference to Schorr et al. 2014.

Page 3, Line 52: The results (e.g. Page 5, line 46 and figure 4) suggest substantial variability between individual, with the range of individual median dive depths spanning almost 600m. In the discussion, individual variability is also highlighted. Based on this, why did you choose to pool your data in order to determine the deep vs shallow dive cutoff? The cutoff applied to one individual with a more limited sample could be substantially biased if it was influenced by a large number of data points from another animal behaving very differently. The scatter plot in figure 2 does not easily allow a reader to assess the degree to which individual data might be influencing pooled metrics, given the overall number of data points displayed. A hard cutoff doesn't deal well with the tails in the distribution of deep and shallow dives per individual (Figures 4 &5). While there is likely only a fraction of dives that may be classified differently, given the emphasis on individual variability in the discussion, the authors should better qualify why/how pooling data for this important cutoff was chosen.

We agree with the reviewer that the pooled scatter plot in Figure 2 does not allow the reader to assess the effect of individual data. To remedy this, we have included a figure in the supplementary materials showing separate scatter plots for each individual and we direct interested readers to this figure. This figure indicates that there is little effect of individual behavior on the 800 m cutoff. It is possible that some points in the tails of the distributions will be miscalculated by using a single

cutoff number, but it is such a small fraction of the overall number of dives that it will not affect population level statistics. Several other studies of *Ziphius* dive behavior utilize a hard cutoff (Tyack et al. 2006, Joyce et al. 2017, Baird et al. 2006, 2008, 2019).

Page 5, line 60. Having referenced the term 'extreme' to previously describe beaked whale behavior, I understand the point- particularly with respect to other species. However, stating that their diving is both extreme and habitual in the same sentence seems a contradiction of terms. At this stage in our understanding of beaked whale diving, the data convincingly suggest that diving to depths greater than 1500m is routine for this species, so I would suggest moving away from representing 'habitual' behavior of beaked whales as extreme behavior.

This is a fair comment, and we agree with the reviewer that it is contradictory to state that their diving is both extreme and habitual. We would have changed the wording here to state that these dives are extreme by other mammalian standards but are habitual for *Ziphius*. However, at the suggestion of Reviewer 1, we have removed this and the subsequent paragraph from the manuscript, as they do not present any new findings. We have edited the text later in the discussion to clarify this wording.

Page 6, Line 17-24. This section is a bit hard to follow and understand. Overall, the paragraph reads as if focused on deep dive interval (IDDI) as a function of recovery time. The third sentence states that pilot whales average 8-12 minutes between foraging dives (i.e. a short IDDI) but then states that pilot whales spend long periods near the surface between foraging dives. That time should be included in the IDDI, so if that is 8-12 minutes, pilot whales would appear to spend (proportionately) much more time on deep foraging dives, not "a comparable amount over an entire diel cycle". Ultimately, this section seems to confound IDDI, foraging rates, surface intervals (as defined by tags in this study), and time near the surface (as referenced for other species) across species, and could stand to be rewritten to clarify the underlying point.

We agree with the reviewer that this paragraph was unwieldy and could have used clarification. Given this comment and the suggestion of Reviewer 1, we have removed this paragraph from the manuscript (see previous comment).

Page 6, line 60 to page 7 line 6. This section should reference Schorr et al. (2014), Tyack et al. (2015) and Schorr et al. (2015), all of which explore this point in detail for Cuvier's beaked whales.

References not yet cited are:

Tyack PL, Calambokidis J, Friedlaender AS, Goldbogen JA (2015) Formal Comment on Schorr GS, Falcone EA, Moretti DJ, Andrews RD (2014) First Long-Term Behavioral Records from Cuvier's Beaked Whales (*Ziphius cavirostris*) Reveal Record-Breaking Dives (BL Southall, Ed.). PLoS ONE:1–4

Schorr GS, Falcone EA, Moretti DJ, Andrews RD (2015) Rebuttal to the Formal

Comment on Schorr et al. (2014) submitted by Tyack et al. (2015) (A Fahlman, Ed.).
PLOS ONE 10:e0142437

Thank you, we have added these references.

Page 7 line 56-58. Suggest including the above references (Tyack et al. 2015 and Schorr et al. 2015) here, as well, given they dealt directly with the issue of short versus long datasets. Also, suggest changing the word from 'Our study INDICATES...' to 'Our study CONFIRMS...' since you reference a previous study having described this.

We have added those references and changed the wording of that sentence. In addition, Reviewer 1 requested that that paragraph be re-written, and we direct the reviewer to the text for those edits.

Page 8, line 20-23: In addition to this unpublished manuscript, it seems like you should also reference a paper demonstrating how data gaps can be reduced by utilizing receivers other than just those on orbiting satellites. Suggest adding after "...incomplete tag records..." 'unless shore-based receivers are utilized (Jeanniard-du-Dot et al., 2017)', or similar. Also, if the journal has issues with references that can't be accessed in any way at the time of publication, in addition to the unpublished manuscript you might also reference a gray literature article dealing specifically with this issue.

Thank you for this suggestion. We have added the sentence "...incomplete tag records. In areas where animal occur close to land, shore-based receivers can improve data retrieval (Jeanniard-du-Dot et al. 2017)." As a side note, we would like to be able to use shore-based receivers to reduce these data gaps but, unfortunately, the topography of our study area makes this impossible.

We prefer to retain the Quick et al. reference as it addresses the same type of tag (whereas the Schorr et al. 2017 gray article discusses a Fastloc GPS tag) and study location. Since the time of original submission of this article, the Quick et al. paper has been accepted for publication in *Animal Biotelemetry* and will therefore be listed as 'in press'.

Jeanniard-du-Dot T, Holland K, Schorr GS, Vo D (2017) Motes enhance data recovery from satellite-relayed biologgers and can facilitate collaborative research into marine habitat utilisation. *Animal Biotelemetry* 5:1–15

Schorr GS, Rone BK, Falcone Erin A (2017) Integrated measurement of Naval sonar operations and precise cetacean locations: Integration of Fastloc GPS into a LIMPET tag. Final report for Task C: Contrac No. N66604-14-C-2438. Availalbe at www.marecotel.org

Page 8 line 24: I agree that sensor issues have not always been addressed in detail in previous peer-reviewed literature, and I commend the authors for more fully discussing this issue and how it was dealt with. However in regards to data gaps and

other limitations of the tags not having been acknowledged, several studies on beaked whales using LIMPET tags referenced in your paper have acknowledged/addressed at least some of the limitations of these tags (e.g. the presence and influence of data gaps, the limitations of summarized data, etc.), at least to some degree (see Schorr et al., 2014 & 2015, Joyce et al. 2016, Jeanniard-du-Dot et al., 2017). This statement should identify specific data issues that have not been adequately addressed, and/or these references should be added.
Thank you, we have added these references.

Page 8 line 27: Clarify how your data identifies the need for tags to transmit data “more quickly”, since this doesn’t seem clear from the above. Is this need based on Argos message length or size (e.g. faster transmission would allow for transmission of longer or larger messages)? Also, stating there is a need for tags capable of “handling the extraordinary diving capabilities” suggests that the tags you and previous authors are using are not. As this is the final sentence of the discussion, it should be more clearly supported by the data discussed above.

We have replaced this sentence with the following: “Our analysis of the errors associated with these tags has indicated several areas where satellite tags could be improved. Given the extraordinary diving capabilities of these animals, there is a need to develop tags with sensors that are calibrated and rated to survive the extreme depths that these animals are capable of diving. To avoid losing information to data gaps from messages not fully transmitting at the surface, there is also a need for the tags to be able to contact a satellite and push their messages through more quickly to account for the short surface durations of these animals. Developing tags with these capabilities will greatly improve our knowledge of the world’s deepest mammalian divers.”

The main issue we have had to deal with, other than apparent pressure sensor failures, has been the existence of numerous gaps in the data record from each tagged whale. Given the very short surface durations of these beaked whales, we cannot push longer or larger messages, but we could reduce these gaps if there was a quicker connection and transmission time at each surfacing.

We don’t intend to suggest that tags that we or previous authors have used have inaccurately described the behavior of beaked whales, but we do believe that current tags are not ideally suited to studying beaked whales. A tag specifically designed for beaked whales would allow for a full representation of their behavior, including possibly very deep (>3,000 m) dives, while maximizing the data returned from each tag, and with fewer tag failures.

Page 8 line 42. Following the previous comment, again what exactly is meant by ‘...develop tags capable of accurately measuring behavior at the extreme depths at which these animals forage’? Are you referring only to the need for a LIMPET tag with a deeper rated pressure sensor? Suggest re-wording this sentence since you describe all of the deep dives in your dataset as extreme, so this could be read to

suggest that the tag used here (or any other tag currently available) is not capable of 'accurately' measuring behavior.

We agree that this wording was problematic. We have changed that sentence to read: "Finally, there is a need for beaked whale-specific tags that are calibrated to the remarkable depths at which these animals routinely dive."

Page 22, line 30. You state the 800 m cutoff has been used "previously" to segregate deep and shallow dives classes, but this does not appear to be referenced elsewhere in the manuscript. See also previous comment about using this cutoff limit. Please add a reference or clarify.

We thank the reviewer for this catch and have added the appropriate references (Baird et al. 2006, 2008).